# *CLEF*: CLINICALLY-GUIDED CONTRASTIVE LEARNING FOR ELECTROCARDIOGRAM FOUNDATION MODELS

## ABSTRACT

The electrocardiogram (ECG) is a key diagnostic tool in cardiovascular health. Single-lead ECG recording is integrated into both clinical-grade and consumer wearables. While self-supervised pretraining of foundation models on unlabeled ECGs improves diagnostic performance, existing approaches do not incorporate domain knowledge from clinical metadata. We introduce a novel contrastive approach that utilizes an established clinical risk score to adaptively weight negative pairs: *clinically-guided contrastive learning*. It aligns the similarities of ECG embeddings to clinically meaningful differences between subjects, with an explicit mechanism to handle missing metadata. Using 12-lead ECGs from 161K patients in MIMIC-IV dataset, we pretrain single-lead *ECG foundation models* at three scales, collectively called *CLEF*, using only routinely-collected metadata without requiring per-sample ECG annotations. We evaluate *CLEF* on 18 clinical classification and regression tasks across 7 held-out datasets, and benchmark against 5 foundation model baselines and 3 self-supervised algorithms. When pretrained on 12-lead ECG data and tested on lead-I data, *CLEF* outperforms self-supervised foundation model baselines: the medium-sized *CLEF* achieves average AUROC improvements of at least $2.6\%$ in classification and average reductions in MAEs of at least $3.2\%$ in regression. Comparing with existing self-supervised learning algorithms, *CLEF* improves the average AUROC by at least $1.8\%$. Moreover, when pretrained only on lead-I data for classification tasks, *CLEF* performs comparably to the state-of-the-art ECGFounder, which has been trained in a supervised manner. Overall, *CLEF* allows more accurate and scalable single-lead ECG analysis, advancing remote health monitoring. We will publish our code and pretrained *CLEF* models.

## 1 INTRODUCTION

The electrocardiogram (ECG) (Geselowitz, 1989) captures the heart's electrical activity as a sequence of voltage fluctuations (Lilly, 2012). While ECG interpretation is a clinical task, ECG-based AI started matching clinician performance, and enabled large-scale applications such as detecting heart conditions from wearables. Furthermore, Yao et al. (2021) found out that clinicians using AI were 30% more accurate in diagnosing left ventricular dysfunction from ECGs than those without AI support. The rise of wearables and mobile health devices with single-lead ECGs enables continuous monitoring beyond clinical settings, supporting early cardiac event detection, long-term tracking, and proactive interventions. Nevertheless, accurate analysis of single-lead ECGs remains a challenge due to the reduced spatial information and the increased noise and motion artifacts (Khamis et al., 2016; Halvaei et al., 2021; Khunte et al., 2023). Prior work has shown that single-lead ECG devices can be used to diagnose atrial fibrillation Svennberg et al. (2015), and that spatial correlations among ECG leads can help infer missing leads (Nelwan et al., 2004). Single-lead ECG is used for practical diagnoses, especially in wearable contexts (Qin et al., 2023; Li et al., 2024).

Collecting sufficiently large, human-annotated, single-lead ECGs is impractical, given the variability in downstream tasks across devices and health conditions. ECGs from healthcare organizations are usually provided in the standard 12-lead format but contain limited annotations. A more practical solution is to pretrain on 12-lead ECGs to learn generalizable representations, then fine-tune on smaller, task-specific data curated for the target health condition. Often, pretraining is performed using *contrastive learning* with data augmentation (Chen et al., 2020; Soltanieh et al., 2022), encouraging the representations of similar samples to be as close to one another while pushing apart those of

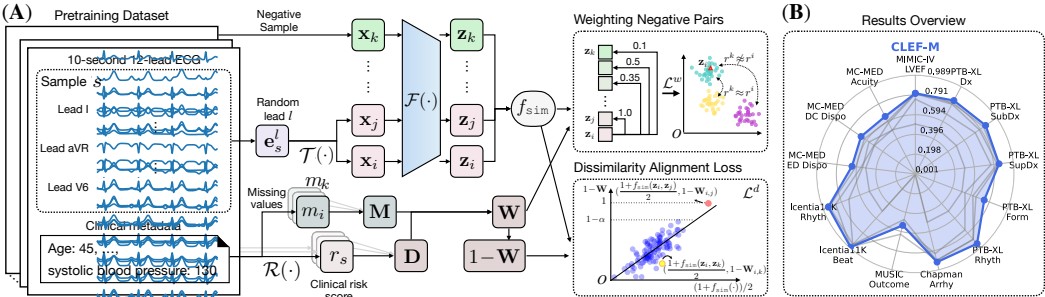

Figure 1: *CLEF*'s framework and performance overview (see §2 for notations). **(A)** Our clinically-guided contrastive pretraining. Key components include a negative weighting loss $\mathcal{L}^w$ and a dissimilarity alignment loss $\mathcal{L}^d$ that work in tandem to guide contrastive learning with clinical knowledge. **(B)** Spider plot on AUROC performance of *CLEF*-M (our medium-sized model) across 13 downstream classification tasks. Baseline performances are in gray lines (see §4).

dissimilar ones. This not only uses information from within samples (e.g. data augmentations), but also uses contextual factors, which can provide important cues for similarity that are not directly observable in the input. Contrastive learning for images is guided by the spatial proximity of image viewpoints or content (Thoma et al., 2020a;b), or by sample identities (Haslum et al., 2024). Prior work also incorporates subject and signal attributes in contrastive learning (Kim et al., 2025), but did not incorporate clinical knowledge, nor handle missing metadata.

Metadata recorded alongside ECGs, such as patient demographics, provides a potential source of contextual data. Some use metadata as model inputs (Erturk et al., 2025), while others have incorporated metadata prediction as part of the pretraining objective (Wang et al., 2024), or as downstream prediction targets (e.g., predicting gender or age from ECGs (Li et al., 2024)). Since gender or age is not a primary objective, this merely demonstrates the model's ability to distinguish between patient groups. In this paper, we take a more integrative strategy to employ metadata as contextual information in contrastive pretraining, since different people might be associated with different risks of a particular health outcome. We leverage metadata in the form of *clinically validated risk scores* to pretrain a foundation model (FM) for single-lead ECGs. Risk scores estimate a person's risk of experiencing an adverse health outcome in a specified time period, such as risk of developing cardiovascular disease in the next ten years (Conroy et al., 2003; working group and risk collaboration, 2021a;b). They take various metadata variables as inputs (e.g., patient demographics, past medical history, and comorbidities), and output a quantitative assessment of the risk of adverse outcomes. Risk scores can guide clinical decisions Hughes et al. (2023), such as whether to prescribe a drug (Lip et al., 2010), or perform invasive procedures (Fox et al., 2006). Our **contributions** are:

1. **Clinically-guided contrastive learning for ECG representation learning.** We leverage metadata to calculate clinically informative risk scores that provide flexible similarity relationships between unlabeled ECG samples. Unlike classic contrastive approaches that rely on binary notions of similarity/dissimilarity, our method incorporates clinical risk scores as soft guidance, enabling the model to learn richer and more nuanced representations.

2. **Extensive empirical analysis and benchmarking for ECG FMs.** We: (i) evaluate downstream performance on lead I/II ECG classification and regression, (ii) compare against 12-lead ECG approaches, (iii) assess robustness across different model architectures and pretraining methods, (iv) perform ablation studies on loss components and handling missing metadata, and (v) use linear probing analysis to assess representation quality.

3. **Effectiveness across diverse downstream tasks.** We evaluate on a wide range of applications. *CLEF* consistently consistently outperforms strong baselines, achieving an average improvement of 3.1% in classification AUROC and a reduction of 2.9% in regression MAE.

## 2 CLINICALLY-GUIDED CONTRASTIVE PRETRAINING

Let $\mathbb{D} = \{(\mathbf{e}_s = (\mathbf{e}_s^1, \ldots, \mathbf{e}_s^{12}), \mathbf{a}_s)\}_{s=1}^N$ denote the *pretraining dataset*, where $\mathbf{e}_s^l \in \mathbb{R}^t$, of length $t$, represents the ECG signal from lead $l$ of the $s$-th sample, $\mathbf{a}_s = \{\mathbf{a}_s^1, \ldots, \mathbf{a}_s^A\}$ denotes demographic or clinical *metadata* (i.e., attributes like age or blood pressure) associated with the subject and

not labels specific to the current ECG recording, and $N$ is the number of 12-lead ECG samples. Importantly, certain attributes might be missing in some samples' metadata. Let $\mathcal{R}$ denote a standard, clinically-validated function that combines a subject's metadata into a numerical estimate of the likelihood of a future health outcome (e.g. sudden cardiac arrest). We define $r_s = \mathcal{R}(\mathtt{a}_s)$ as a *risk score* associated to the sample $\mathbf{e}_s$. Our objective is to train a single-lead ECG FM $\mathcal{F}(\cdot)$ on $\mathbb{D}$ such that the geometry of the *embedding* space, i.e., outputs $\mathbf{z} = \mathcal{F}(\mathbf{e}_s^l) \in \mathbb{R}^h$, with a dimension of $h$, reflects clinically meaningful similarities according to the risk score function $\mathcal{R}(\cdot)$.

To measure the success of our objective, we attach a linear layer $\mathcal{G}(\cdot)$ on top of the embeddings of condition-specific single-lead ECGs produced by $\mathcal{F}(\cdot)$. Let $\mathbb{C} = \{(\mathbf{e}_s^l, y_s)\}_{s=1}^M$ denote a *downstream evaluation dataset*, where $\mathbf{e}_s^l \in \mathbb{R}^t$, represents the ECG signal from lead $l$, and $y_s$ indicates the ground-truth label associated to that ECG signal. Note that the lead is fixed across all samples in the test set (e.g. lead I). We evaluate $\mathcal{F}(\cdot)$ under two downstream scenarios: (1) *fine-tuning*, where all parameters of $\mathcal{F}(\cdot)$ and $\mathcal{G}(\cdot)$ are updated during training on the training split of $\mathbb{C}$, and (2) *linear probing*, where the pretrained $\mathcal{F}(\cdot)$ is kept frozen and only the linear head $\mathcal{G}(\cdot)$ is updated to map the embeddings $\mathbf{z} = \mathcal{F}(\mathbf{e}_s^l)$ to task-specific labels. To maintain a fair comparison, we adopt a consistent set of hyperparameters across all experiments; further details are provided in Appendix C.1.

For binary labels $y \in \{0, 1\}$, we report the AUROC metric, which measures the model's ability to discriminate between the positive and negative classes by computing the area under the Receiver Operating Characteristic (ROC) curve, representing the trade-off between true positive rate and false positive rate across all possible classification thresholds. For categorical labels $y \in \{1, \cdots, K\}$ where $K$ is the number of classes, we report the macro AUROC score using the one-vs-rest extension, where each class is compared against all others and the resulting scores are macro-averaged (i.e. unweighted mean). For numerical labels $y \in \mathbb{R}_0$, we report the mean absolute error (MAE) between the predicted and true values, providing an interpretable metric of prediction accuracy in the same units as the target variable. Performance is summarized as a percentage improvement over baseline.

## 2.1 CONTRASTIVE LEARNING

*Contrastive Loss.* We revisit contrastive learning (Chen et al., 2020). Given a sample $\mathbf{e}_s^l \overset{\text{i.i.d.}}{\sim} \mathbf{e}_s \overset{\text{i.i.d.}}{\sim} \mathbb{D}$, an independent stochastic transformation $\mathcal{T}(\cdot)$ is applied to $\mathbf{e}_s^l$ to obtain an augmented *view*, denoted as $\mathbf{x}_i = \mathcal{T}(\mathbf{e}_s^l)$. Similarly, we generate another view $\mathbf{x}_j = \mathcal{T}(\mathbf{e}_s^l)$ forming a *positive pair* $(\mathbf{x}_i, \mathbf{x}_j)$ in contrastive learning. A batch of $B$ samples yields $2B$ augmented views, such that any $k \in \{1, 2, \ldots, 2B\} \setminus \{i, j\}$ forms *negative pairs* with both $\mathbf{x}_i$ and $\mathbf{x}_j$, i.e., $(\mathbf{x}_i, \mathbf{x}_k)$ and $(\mathbf{x}_j, \mathbf{x}_k)$. Using a model $\mathcal{F}(\cdot)$, the augmented views are encoded into embeddings denoted as $\mathbf{z}_i, \mathbf{z}_j$, and $\mathbf{z}_k \in \mathbb{R}^h$. For each pair of $(\mathbf{x}_i, \mathbf{x}_j)$, the classic contrastive loss, termed NT-Xent (Chen et al., 2020), is given by

$$\mathcal{L}_{i,j} = -\log \frac{\exp\left(f_{\mathtt{sim}}\left(\mathbf{z}_i, \mathbf{z}_j\right)/\tau\right)}{\sum_{k=1}^{2B} \mathbb{1}_{[k \neq i]} \exp\left(f_{\mathtt{sim}}\left(\mathbf{z}_i, \mathbf{z}_k\right)/\tau\right)} \ , \tag{1}$$

where $f_{\mathtt{sim}}(\mathbf{z}_i, \mathbf{z}_j) = \frac{\mathbf{z}_i^\top \mathbf{z}_j}{||\mathbf{z}_i|| \cdot ||\mathbf{z}_j||} \in [-1, 1]$ is cosine similarity, and $\tau$ is a temperature hyperparameter.

*Weighted Contrastive Loss.* While Eq. (1) treats all negative pairs equally, recent works highlight that negative pairs could contribute unequally to representation learning Robinson et al. (2021); Li et al. (2023); Zhuang et al. (2024); Yang et al. (2024). Conceptually, in the embedding space, negative samples are pushed away from a positive sample according to their similarity. Specifically, the less similar a negative sample is to the positive, the more it is pushed away. By introducing a weighting term $\mathbf{W}_{ik}$ for each pair of negative embeddings $(\mathbf{z}_i, \mathbf{z}_k)$, we can adjust the relative importance of negatives according to their similarity to the positive, having

$$\mathcal{L}_{i,j}^w = -\log \frac{\exp\left(f_{\mathtt{sim}}\left(\mathbf{z}_i, \mathbf{z}_j\right)/\tau\right)}{\sum_{k=1}^{2B} \mathbb{1}_{[k \neq i]} \mathbf{W}_{ik} \exp\left(f_{\mathtt{sim}}\left(\mathbf{z}_i, \mathbf{z}_k\right)/\tau\right)} \ . \tag{2}$$

Eq. (2) generalizes Eq. (1) by replacing the uniform treatment of negatives with an adaptive weighting.

## 2.2 CLINICAL METADATA

*Cardiovascular Risk Score.* A risk score, quantified on clinical and demographic attributes, is the likelihood of experiencing an adverse outcome. They translate complex physiological factors into

interpretable values that guide clinical actions and interventions. The Systematic Coronary Risk Evaluation 2 score (SCORE2) commonly used by healthcare professionals, estimates the 10-year risk of cardiovascular disease events (death, myocardial infarction, or non-fatal stroke). SCORE2 is obtained based on 7 variables: age ($a_s^1$), gender ($a_s^2$), smoking status ($a_s^3$), systolic blood pressure (SBP, $a_s^4$), diabetes status ($a_s^5$), total cholesterol ($a_s^6$), and high-density lipoprotein cholesterol ($a_s^7$). Let $\mathbf{u}_s = [u_s^1, \cdots, u_s^6] = g(\mathbf{a}_s)$ denote the vector after standardization of the ordered metadata feature $\mathbf{a}_s = [a_s^1, a_s^3, \ldots, a_s^7]$. For each sample $\mathbf{e}_s$, we assign a risk score $r_s \in [0, 1]$, defined by the risk score function $\mathcal{R}(\cdot)$ as that SCORE2 value of $\mathbf{e}_s$:

$$r_s \stackrel{\text{def}}{\equiv} \mathcal{R}(a_s) = 1 - S_0(t)^{\exp(\mathbf{b}_1 \mathbf{u}_s^\top + u_s^1 \mathbf{b}_2 \mathbf{u}_s^\top)}. \tag{3}$$

where $S_0(t)$ is the baseline survival [1] function at year $t$, and $\mathbf{b}_1, \mathbf{b}_2$ are the variable's coefficient estimated from the European cohort data covering over 600K individuals with more than 30K cardiovascular events (values varying by age and gender). Given that augmentations do not change metadata, for $\mathbf{x}_i, \mathbf{x}_j$ of sample $s$, we have $r^i = r^j = r_s$ ($r^i, r^j$ are risk scores for $\mathbf{x}_i, \mathbf{x}_j$). We provide further details on calculating SCORE2 in Appendix C.6. The *CLEF* framework supports any validated risk score. While we use SCORE2, we note that its development on European populations might limit its generalisability, and the optimal score may differ by application (see Appendix F).

*Handling Missing Metadata.* MIMIC-IV-ECG (Gow et al., 2023) contains 161,352 unique subjects, each with multiple 12-lead ECG recordings. For each subject, the dataset reports only three variables: age, gender, and systolic blood pressure. This is a common challenge in real-world settings, where certain metadata variables are often unavailable either at a specific institution or for particular patients. Such variables are typically collected independently rather than at the time of the ECG recording. To address this issue, we assign different *multipliers* to negative pairs $(\mathbf{x}_i, \mathbf{x}_k)$ based on the number of metadata variables available when calculating their risk scores $(r^i, r^k)$. Specifically, we calculate

$$\mathbf{M}_{ik} = \exp\left(-\frac{A - m_i}{A} \times \frac{A - m_k}{A}\right), \tag{4}$$

where $A$ is the number of variables (e.g., $A = 7$ in SCORE2), and $m_i/m_k$ are the number of missing variables for $\mathbf{x}_i/\mathbf{x}_k$. Thus, $\mathbf{M}_{ik} \in (0, 1]$ indicates the level of missing metadata for a pair $(\mathbf{x}_i, \mathbf{x}_k)$, which functions as a relative reliability adjustment to ensure that only well-supported risk differences meaningfully shape the embedding geometry, we further elaborate this in section 2.3.

## 2.3 GUIDING REPRESENTATION LEARNING WITH CLINICAL RISK SCORES

*Risk Score Dissimilarity.* We aim to enhance weighted contrastive loss in Eq. (2) by using risk scores in Eq. (3) for pretraining our ECG FM $\mathcal{F}(\cdot)$. Our objective is to guide $\mathcal{F}(\cdot)$ toward learning embeddings that capture clinically relevant patterns, building a latent space for ECG signals where the distance between embeddings reflects dissimilarities in risk scores. For a negative pair $(\mathbf{x}_i, \mathbf{x}_k)$ with corresponding $(r^i, r^k)$, let $\delta_{ik} = (r^i - r^k)^2$. We define a negative pair's *dissimilarity*:

$$\mathbf{D}_{i,k} = (1 - \alpha)\frac{\delta_{ik} - \delta^{min}}{\delta^{max} - \delta^{min}} + \alpha, \quad \text{where} \quad \delta^{max} = \max_{i,k}(\delta_{ik}) \quad \text{and} \quad \delta^{min} = \min_{i,k}(\delta_{ik}). \tag{5}$$

Parameter $\alpha \in [0, 1]$ controls the minimum distance between the positive pair and all negative pairs. Dissimilarity to all negative pairs stays within $[\alpha, 1]$, having $\mathbf{D}_{i,j} = 0$ only for the positive pair. For $(\mathbf{x}_i, \mathbf{x}_k)$, even when their metadata indicates $r^i = r^k$, we have their dissimilarity $\mathbf{D}_{i,k}$ at least $\alpha$.

*Weighting Negative Pairs.* By combining Eq. (4) and (5) via the Hadamard product, we define the *weight matrix* $\mathbf{W} = \mathbf{D} \odot \mathbf{M}$, where each entry is given by $\mathbf{W}_{ik} = \mathbf{D}_{i,k} \cdot \mathbf{M}_{i,k} \in [0, 1]$. This value represents the weight assigned to pushing the embedding of $\mathbf{x}_k$ away from that of $\mathbf{x}_i$. Intuitively, the weight increases when the risk scores of $\mathbf{x}_i$ and $\mathbf{x}_k$ differ more; conversely, it decreases when the risk scores are similar. Importantly, when many metadata variables are missing, and SCORE2 would rely heavily on default imputations (see Appendix C.6), the resulting risk differences are thus less trustworthy. In such cases, the corresponding $\mathbf{M}_{ik}$ between the negative pairs moves the weighting closer to a uniform SimCLR-like behavior, preventing uncertain or noisy risk differences from exerting undue influence on the contrastive objective. For each batch of data, we calculate our *clinically-guided contrastive loss* by

---

[1]The probability of not experiencing a cardiovascular event over 10 years for a reference individual.

$$\mathcal{L}^w = -\frac{1}{2B} \sum_{i=1}^{2B} \mathcal{L}_{i,j}^w = -\frac{1}{2B} \sum_{i=1}^{2B} \log \frac{\exp\left(f_{\texttt{sim}}\left(\mathbf{z}_i, \mathbf{z}_j\right)/\tau\right)}{\sum_{k=1}^{2B} \mathbb{1}_{[k \neq i]} \mathbf{W}_{ik} \exp\left(f_{\texttt{sim}}\left(\mathbf{z}_i, \mathbf{z}_k\right)/\tau\right)} \quad . \tag{6}$$

*Dissimilarity Alignment Loss.* We compute the mean squared error loss between the cosine similarity of each embedding and its corresponding weight, averaged across all pairs in the batch.

$$\mathcal{L}^d = \frac{1}{B^2} \sum_{i,j} \left( \frac{1 + f_{\texttt{sim}}(\mathbf{z}_i, \mathbf{z}_j)}{2} - (1 - \mathbf{W}_{ij}) \right)^2 \quad . \tag{7}$$

Note that cosine similarity $f_{\texttt{sim}} \in [-1, 1]$ is rescaled to $[0, 1]$. This encourages the model to map clinically similar ECG signals to nearby embeddings, while pushing dissimilar signals apart. This supports clinical use, where models must highlight differences between ECG signals to distinguish clinical categories (e.g., diagnoses or prognoses). Our final objective is: $\mathcal{L} = \mathcal{L}^w + \mathcal{L}^d$, where, in this paper, the contribution of the two losses are considered equal (details in Appendix D.6).

## 3 EXPERIMENTAL EVALUATION

We evaluate against (i) existing FMs and models pretrained on ECG data, (ii) widely-used self-supervised learning algorithms, and (iii) a state-of-the-art (SOTA) supervised foundation model.

**Foundation Model $\mathcal{F}$.** We use ResNeXt1D (Hong et al., 2020)[2], built on ResNeXt Xie et al. (2017), using one-dimensional convolutional filters, and widely used as a benchmark for ECG processing Li et al. (2024). We evaluate three model configurations: small (448K parameters), medium (30.7M parameters), and large (296M parameters), which differ in network depth, width, and complexity. A more detailed model structure description can be found in Appendix C.3 and Table S7.

**Pretraining Dataset $\mathbb{D}$.** We train $\mathcal{F}(\cdot)$ using the MIMIC-IV-ECG dataset (Gow et al., 2023), which contains $161,352$ unique patients, each with multiple ECG recordings. To ensure that signals from the same patient are not considered as negative pairs during model pretraining, we use the first ECG recording of each patient. Each ECG record $\mathbf{e}$ is sampled at 500Hz over 10 seconds, resulting in a sequence length of $t = 5,000$. Following the common practice, we apply a Butterworth bandpass filter between 0.67 and 40 Hz, followed by z-score normalization for each sample.

**Stochastic Data Augmentation $\mathcal{T}$.** To simulate real-world signal perturbations, we apply some noise derived from free-living[3] ECG recordings, including (i) muscle noise, (ii) movement artifacts, (iii) baseline wander, (iv) white noise, or (v) no perturbations. Function $\mathcal{T}$ is a random selection from these five choices, each with equal probability of $p = 0.2$. For details, see Appendix C.2.

**Random Lead Selection.** Our *CLEF* can be fine-tuned to any lead, which is crucial since devices and health-monitoring scenarios vary. Also, many wearables only approximate a lead; e.g., smartwatches approximate lead I, and the target lead may not be known at pretraining time. With this motivation, we do not restrict training to a specific lead as pretraining exclusively on a specific lead (explored in section 4, Figure 4) requires one separate model per lead and is less flexible for cross-device use. Instead, for each sample in a batch, we randomly select one of the 12 leads and apply stochastic augmentation, thus every batch contains diverse samples from multiple leads. In this way, the model is collectively trained across all 12 leads, allowing $\mathcal{F}$ to generalize and later be fine-tuned to any lead.

**Downstream Datasets $\mathbb{C}$.** We evaluate on various downstream tasks from cardiovascular conditions, including 3 well-established benchmarks used in (Li et al., 2024; Na et al., 2024), and 4 newly curated datasets from wearables or emergency visits. Tasks include multi-label diagnostic (e.g. heart block, myocardial infarction, hypertrophy), form (i.e. wave morphology), and rhythm statements on PTB-XL dataset (Wagner et al., 2020), left ventricular ejection fraction (LVEF) regression and classification at 50% threshold on MIMIC-IV-ECG dataset (Gow et al., 2023), and multi-label disease classification on the Chapman dataset (Zheng et al., 2020). We also include MC-MED (Kansal et al., 2025) and Aurora BP (Mieloszyk et al., 2022) for blood pressure estimation, the MUSIC dataset (Martin-Yebra et al., 2025) for long-term cardiovascular outcomes such as sudden cardiac death (SCD), and

---

[2]The PyTorch implementation of the model is adapted from github.com/hsd1503/resnet1d.

[3]The noise data is available at: physionet.org/content/ecg-ppg-simulator-arrhythmia.

Table 1: Results for finetuning on lead I (left) and lead II (right) ECGs. AUROCs are reported for 7 clinical tasks: LVEF refers to left ventricular ejection fraction; Dx denotes diagnosis; SubDx denotes subdiagnosis; SupDx denotes superdiagnosis; Form refers to waveform morphology; Rhyth denotes rhythm; Arrhy denotes arrhythmia. Best result **bolded**. The second best underlined.

| Dataset / Task | Lead I | | | | | | | Lead II | | | | | | |
|---|---|---|---|---|---|---|---|---|---|---|---|---|---|---|
| | MIMIC-IV LVEF | PTB-XL Dx | SubDx | SupDx | Form | Rhyth | Chapman Arrhy | MIMIC-IV LVEF | PTB-XL Dx | SubDx | SupDx | Form | Rhyth | Chapman Arrhy |
| Moirai 2024 (91M) | .4968 | .5003 | .4966 | .5011 | .5158 | .5031 | .4992 | .5000 | .5014 | .5001 | .4993 | .5003 | .4988 | .4987 |
| Moment 2024 (125M) | .7763 | .7780 | .7627 | .7700 | .6322 | .8243 | .8150 | .8024 | .7896 | .7788 | .8260 | .6164 | .8754 | .8905 |
| ST-MEM 2024 (85M) | .7751 | .7763 | .7639 | .7800 | .5724 | .7549 | .8113 | .7499 | .6764 | .7284 | .7637 | .5319 | .7489 | .8228 |
| KED 2024 (8M) | .8330 | .8390 | .8332 | .8302 | .6696 | .8887 | .8897 | .8073 | .8072 | .8304 | .8407 | .6492 | .8736 | .8941 |
| *CLEF*-S (448K) | .8170 | .8268 | .8448 | **.8452** | .6738 | .9304 | .9033 | .8166 | .8205 | .8486 | .8445 | .6913 | .9305 | .9047 |
| *CLEF*-M (30.7M) | .8083 | .8292 | **.8566** | .8430 | .7409 | .9361 | .9061 | .8079 | **.8307** | **.8555** | .8446 | .7378 | .9326 | **.9089** |
| *CLEF*-L (296M) | .7858 | **.8472** | .8397 | .8273 | .7162 | .9325 | .9010 | .8194 | .8193 | .8438 | .8409 | .7478 | **.9512** | .9011 |
| ECGFounder 2024 (76.3M) | **.8512** | .8457 | .8500 | .8376 | **.7626** | **.9501** | **.9090** | **.8312** | .8024 | .8244 | .8437 | .6823 | .9229 | .8755 |

Icentia11K (Tan et al., 2019), a large-scale continuous wearable ECG dataset supporting beat and rhythm classification. More details are provided in Appendix B and C.1, respectively.

**Foundation model baselines.** (1) ST-MEM (Na et al., 2024): a spatio-temporal masked autoencoder trained to reconstruct randomly masked ECGs. (2) KED (Tian et al., 2024): that aligns ECG signals with textual reports, enabling joint ECG–text representations. (3) Moirai (Woo et al., 2024): a forecasting model for time series, supporting both univariate and multivariate prediction. (4) Moment (Goswami et al., 2024): a transformer-based model designed for univariate time series tasks.

**Self-supervised learning baselines**. (1) SimCLR (Chen et al., 2020): a canonical contrastive learning framework, as described in §2, Eq. (1). (2) BYOL (Grill et al., 2020): an augmentation-based approach that eliminates negative pairs by training another network to predict the target network's representation of the same signal. (3) MoCo (He et al., 2020): maintains a dynamic memory bank to store representations from past batches, reducing the reliance on large in-batch negatives.

**Supervised baseline.** Concurrent to this work, Harvard–Emory researchers released the HEEDB dataset (Ghanta et al., 2025) of 12-lead ECG recordings with 10M ECGs from 1.8M patients annotated with 150 diagnostic categories. While we could not get access to the dataset, they released a supervised ECG FM, ECGFounder (Li et al., 2024), trained for multi-label classification over the 150 categories. We run ECGFounder on our evaluation datasets, and since it is trained on a large labeled dataset, we consider it as an upper bound relative to semi-supervised methods, including our proposed *CLEF*. Implementation details for all baselines are provided in Appendix C.4 and C.5.

## 4 RESULTS

**Finetuning on Lead I & II (in-clinic datasets).** The AUROC results are detailed in Table 1 (with further details in Appendix D.1). *CLEF* outperforms the best baseline across both leads, with the best-performing *CLEF* variant for each task achieving an average AUROC improvement over the strongest baseline for that task of 3.1% on lead I, and 4.8% on lead II. On lead I, the *CLEF*-S, -M, and -L variants achieve improvements of 1.0%, 2.6%, and 1.3%, respectively, over the best performing baseline (1.6% on average), while on lead II, gains were even stronger, achieving 2.8%, 4.0%, and 4.2% (3.7% average) performance gain. KED was the best-performing baseline in all but one task, and *CLEF*-M shows statistically significant superiority over KED (p = 0.010, by paired t-test). *CLEF* outperforms KED by $\geq 1.0\%$ on lead I and $\geq 2.8\%$ on lead II across all 3 variants, demonstrating its strong potential for pretraining. Additionally, the averaged confidence interval across all *CLEF* variants and tasks is $\pm 0.01$, showcasing *CLEF*'s consistent performance. *CLEF* performs particularly well on pattern identification tasks (form and rhythm tasks of PTB-XL, and the arrhythmia task of Chapman), with its best variant improving AUROC by 5.9% (lead I) and 8.5% (lead II) on average over the strongest baselines. These results suggest that *CLEF* excels at capturing both morphological features of individual beats and long-range rhythm dependencies across the signal.

*CLEF* maintains robust performance across leads, with *CLEF*-S, -M, and -L differing by 0.3%, 0.0%, and 1.4% on average in AUROC across 7 tasks. In contrast, ST-MEM and KED which are specifically pretrained for ECG tasks, favored lead I, with AUROCs on lead II an average of 4.2% and 1.5% lower, respectively. When compared to the supervised model ECGFounder, *CLEF* performs less

well on lead I (AUROCs lower by 2.9%, 1.5%, and 2.7% for *CLEF*-S, -M, and -L, respectively) but better on lead II (AUROC increases by 1.3%, 2.5%, and 2.6%). This is primarily because ECGFounder's performance drops 3.8% on lead II, compared to lead I (on which the single-lead version of ECGFounder was trained (Li et al., 2024)), while *CLEF* maintains performance across leads. Although the unsupervised *CLEF* did not perform as well as this SOTA supervised model on the lead for which that model was trained, it did achieve improved performance on another lead. Notably, our single-lead *CLEF* even outperforms some baselines when those baselines are pretrained and evaluated on 12-lead ECGs. Specifically, *CLEF*-S (evaluated on lead I & II) outperforms ST-MEM on 5 out of 7 tasks (Details are reported in Appendix D.2 and Table S10).

**Finetuning for Single-lead ECG (out-of-clinic datasets).** We use single-lead ECGs from MUSIC, Icentia11k, and AuroraBP collected by wearables, and MC-MED dataset of Emergency Department.

*Classification.* The AUROC results are detailed in Table 2 (for classification tasks) and Table 3 (for regression tasks). In classification, *CLEF* outperforms all semi-supervised baselines, with the best *CLEF* variant improving AUROC by 2.5% on average over the strongest baseline of the task. The best-performing *CLEF* variant for each task achieves an average AUROC improvement over the strongest baseline for that task of 2.5%. Individually, *CLEF*-S, -M, and -L achieve averaged AUROC improvements of 2.8%, 6.7%, and 2.5% over the best baseline (KED). On Icentia11K Beat and Rhythm tasks, the best-performing *CLEF* variant on each task achieves improved AUROC by 1.9% over the best baselines. When looking at each *CLEF* variant, *CLEF*-S, -M, and -L achieve average AUROC improvements of 1.5%, 2.5%, and 3.7% over the best baseline (KED), highlighting the effectiveness of *CLEF* in capturing quality ECG representations for wearable devices. However, Moment outperformed *CLEF* on MUSIC SCD prediction, the only case where a general time series model surpassed an ECG-specific model; likely because SCD risk depends on long-range and subtle temporal patterns in the ECG, making it more of a forecasting problem than a short-term classification.

*Regression.* We evaluate on the MIMIC-IV LVEF task (explored in prior work (Li et al., 2024)), and also systolic and diastolic blood pressure (SBP and DBP) prediction on the MC-MED and AuroraBP datasets. Target labels are first z-scored based on the statistics of the training set, and then at test time, predicted values are transformed back to the original scale of the labels for evaluation. See Appendix C.1 for details of our experimental setup. The mean absolute error (MAE) is reported in Table 3. The best-performing *CLEF* for each task achieves an average MAE reduction in comparison to the strongest baseline for that task of 2.9%. The *CLEF*-S and *CLEF*-M variants outperform all baselines, achieving average MAE reductions of 3.2% compared to the best baseline (ST-MEM). In contrast, *CLEF*-L outperforms 3 of the 4 baselines (yet MAE is 2.3% higher than ST-MEM on average). *CLEF* also outperforms the supervised model ECGFounder, with *CLEF*-S, -M, and -L achieving lower MAEs than ECGFounder by 5.5%, 5.6%, and 0.2% on average.

**Linear probing.** To evaluate the quality of the representations produced by *CLEF*, the parameters of FMs were kept frozen, and a linear classifier was trained on the output embeddings. The AUROCs are summarized in Figure 2 for *CLEF*-M and 4 baselines. Full results including those for the other 2 variants (*CLEF*-S and -L) are provided in Table S11 (Appendix D.3). The best-performing *CLEF* for each task outperforms the strongest baseline for that task of +7.3%. *CLEF*-M and *CLEF*-L both outperform all baselines, with average AUROC improvements of 8.5% and 9.9% respectively over

Table 2: Finetuning using single-lead ECGs. AUROCs are for: SCD outcome prediction, beat and rhythm recognition, disposition after emergency department visit (ED dispo) and after hospital stay (DC dispo), and ED triage acuity. Best result **bolded**. The second best underlined.

| Model | MUSIC | Icentia11K | | MC-MED | | |
|---|---|---|---|---|---|---|
| | SCD | Beat | Rhythm | ED Dispo | DC Dispo | Acuity |
| Moirai | .0106 | .4998 | .4927 | .5004 | .4996 | .4992 |
| Moment | **.6223** | .9764 | .8540 | .5488 | .5042 | .5782 |
| ST-MEM | .5389 | .9452 | .7115 | .5323 | .5645 | .4943 |
| KED | .4701 | .9424 | .9228 | .5992 | .6200 | .5978 |
| *CLEF*-S | .5493 | .9801 | .9135 | .6065 | .6261 | .5650 |
| *CLEF*-M | .5304 | .9792 | .9328 | .6397 | .6607 | .6510 |
| *CLEF*-L | .5545 | .9800 | .9535 | .5805 | .5800 | .5940 |
| ECGFounder | .5420 | **.9822** | **.9723** | **.6572** | **.6710** | **.6690** |

Table 3: ECG regression tasks. MAEs are reported for LVEF prediction (in %) on MIMIC-IV, and systolic and diastolic blood pressure prediction (in mmHg) on both the MC-MED and Aurora BP data sets. Best result **bolded**. The second best underlined.

| Model | MIMIC-IV | Aurora BP | | MC-MED | |
|---|---|---|---|---|---|
| | LVEF | SBP | DBP | SBP | DBP |
| Moirai | 7.405 | 12.587 | 8.261 | 32.881 | 12.704 |
| Moment | 7.277 | 12.707 | 8.150 | 18.623 | 12.576 |
| ST-MEM | 7.149 | 12.707 | 8.157 | 18.623 | 12.552 |
| KED | 8.313 | 11.844 | **8.032** | 18.598 | 12.647 |
| *CLEF*-S | **6.569** | **11.667** | 8.653 | 17.901 | **12.314** |
| *CLEF*-M | 6.805 | 12.110 | 8.099 | **17.880** | 12.318 |
| *CLEF*-L | 7.313 | 12.819 | 8.865 | 18.763 | 12.441 |
| ECGFounder | 7.900 | 13.577 | 8.332 | **17.880** | 12.378 |

Table 4: Comparing self-supervised pretraining methods with AUROCs of single-lead ECG classification. For datasets that originally have 12 leads, lead I was used for analysis. Instances outperformed by *CLEF* are highlighted with cells having a grey background.

| Model | | MIMIC-IV LVEF | PTB-XL Dx | SubDx | SupDx | Form | Rhyth | Chapman Arrhy | MUSIC Outcome | Icentia11K Beat | Rhyth | MC-MED ED Dispo | DC Dispo | Acuity |
|---|---|---|---|---|---|---|---|---|---|---|---|---|---|---|
| MOCO (2020) | S | .8194 | .5378 | .5803 | .7260 | .5300 | .5128 | .7743 | .5477 | .9133 | .4737 | 0.6111 | 0.6075 | 0.5518 |
| | M | .8112 | .7664 | .7885 | .7757 | .5021 | .5163 | .8119 | .4546 | .9764 | .7115 | 0.5926 | 0.5963 | 0.5659 |
| | L | .6994 | .5014 | .4981 | .5060 | .5063 | .4990 | .5258 | .4563 | .8680 | .5697 | 0.5331 | 0.4865 | 0.4781 |
| SimCLR (2020) | S | .8185 | .7923 | .7790 | .8180 | .5560 | .8979 | .8554 | .5485 | .9722 | .9180 | 0.6150 | 0.6376 | 0.6135 |
| | M | .8243 | .8247 | .8352 | .8210 | .7338 | .9248 | .8888 | .5531 | .9756 | .8936 | 0.6380 | 0.6553 | 0.5743 |
| | L | .8192 | .8393 | .8370 | .8279 | .7137 | .9109 | .8945 | .4799 | .9733 | .9087 | 0.5984 | 0.6129 | 0.5988 |
| BYOL (2020) | S | .8147 | .8134 | .8282 | .8259 | .6883 | .9296 | .8789 | .5448 | .9729 | .9065 | 0.6166 | 0.6177 | 0.6188 |
| | M | .8221 | .8421 | .8221 | .8251 | .7215 | .9351 | .8917 | .4713 | .9766 | .8919 | 0.6306 | 0.6447 | 0.5643 |
| | L | .8221 | .8409 | .8378 | .8239 | .7144 | .9257 | .8978 | .4719 | .9819 | .9032 | 0.5515 | 0.5588 | 0.5147 |

the best baseline (KED). However, *CLEF*-S does not outperform KED or Moment, which indicates that the larger *CLEF* models produce better quality representations. Models pretrained specifically for ECG tasks do not always yield better performance. Moment (which *CLEF*-M outperforms by 10.0%) performs better than ST-MEM (which *CLEF* outperforms by 28.1%).

**Comparison with Self-supervised Algorithms.** Our proposed clinically-guided contrastive learning method is compared against 3 widely used self-supervised pretraining methods: SimCLR, BYOL, and MoCo. Each pretraining method is trained on all 3 size variants of *CLEF*. The AUROC results for the comparator methods are presented in Table 4. A total of 100 out of 117 instances were outperformed by *CLEF* (highlighted in gray). Overall, *CLEF* outperforms all 3 self-supervised pretraining methods, with average AUROC improvements across all sizes of backbone models of 29.8%, 1.8%, and 2.3% in comparison to MOCO, SimCLR, and BYOL, respectively.

**Our Clinically-guided Contrastive Loss with Best Semi-supervised Baseline.** We initialize the best-performing baseline for ECG, i.e. KED, with its original pretrained weights and apply our proposed clinically-guided pretraining approach. AUROC results are presented in Figure 3. It can be

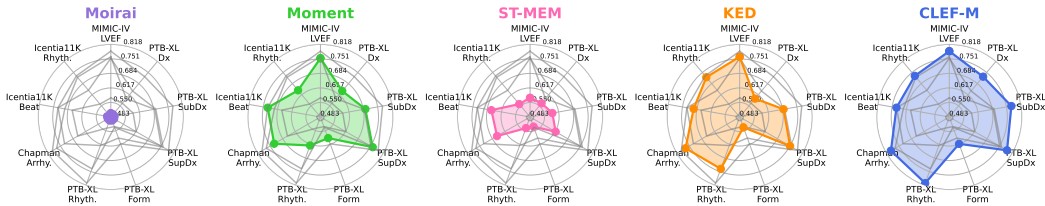

Figure 2: AUROC scores from linear probing on 9 classification tasks, comparing Moirai, Moment, ST-MEM, KED, and our *CLEF*. Each subplot focuses on one model, with others shown in gray for reference. Higher values indicate better performance (see further details in Table S11).

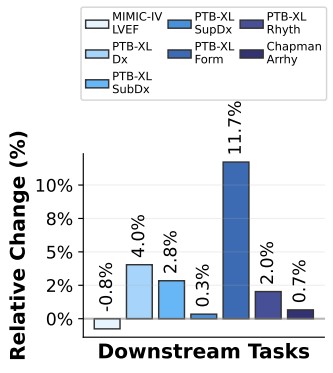

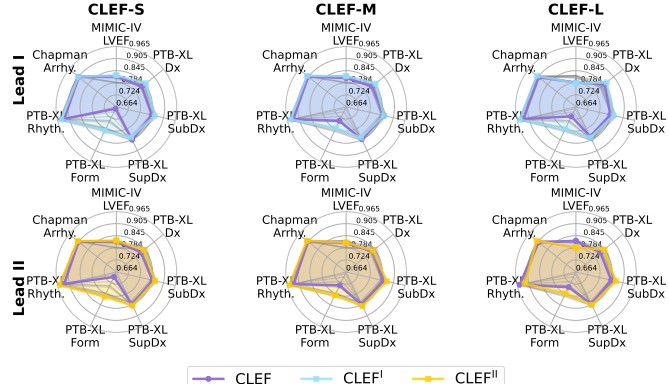

Figure 3: Changes in AUROC of KED after further training with *CLEF* objectives across downstream ECG tasks.

Figure 4: Spider plot comparing *CLEF* with *CLEF* model pretrained on a specific lead (*CLEF*[I] and *CLEF*[II]). AUROC is reported across 3 model variants and 7 downstream tasks.

Table 5: Ablation results on handling missing metadata across *CLEF* models of different sizes, where ¬**M** denotes the model trained without handling the missing metadata. We report both the results from the ablated models, and the performance changes relative to the corresponding *CLEF* models (shown in brackets). The ↑ and ↓ depict performance increase and drop, respectively. For better clarity, we highlighted cases where performance is inferior in blue.

| Task | MIMIC-IV | PTB-XL | | | | | Chapman | MUSIC | Icentia11K | |
|---|---|---|---|---|---|---|---|---|---|---|
| | LVEF | Dx | SubDx | SupDx | Form | Rhyth | Arrhy | Outcome | Beat | Rhyth |
| *CLEF*-S ¬**M** | .8245 (↑ .01) | .8152 (↓ .01) | .8266 (↓ .02) | .8210 (↓ .03) | .6810 (↑ .01) | .9362 (↑ .01) | .8786 (↓ .03) | .4763 (↓ .13) | .9637 (↓ .02) | .9052 (↓ .01) |
| *CLEF*-M ¬**M** | .8191 (↑ .01) | .8268 (↓ .00) | .8311 (↓ .03) | .8245 (↓ .02) | .7252 (↓ .02) | .9230 (↓ .01) | .8839 (↓ .02) | .5513 (↑ .04) | .9749 (↓ .00) | .8856 (↓ .05) |
| *CLEF*-L ¬**M** | .7786 (↓ .01) | .8299 (↓ .02) | .8502 (↑ .01) | .8257 (↓ .02) | .7024 (↓ .02) | .9201 (↓ .01) | .8939 (↓ .01) | .4966 (↓ .10) | .9757 (↓ .00) | .8835 (↓ .07) |

observed that, apart from the LVEF classification task in MIMIC-IV, all tasks gained improvement, on average 3.0% higher AUROC. The form task in PTB-XL gets the greatest improvement (11.7%). More results on pretraining other ECG baseline models are provided in Appendix D.4.

**Pretraining *CLEF* on a Specific Lead.** To assess the potential upper bound of *CLEF*, we pretrain models exclusively on the corresponding lead of the downstream tasks (lead I or II). Pretraining was performed for 10 epochs, with the other downstream finetuning hyperparameters the same as the previous experiments. Results are summarized in Figure 4, and provided in full in Appendix D.5. On lead I, pretraining achieves average AUROC improvements over pretraining using all 12 leads of 3.4%, 1.4%, and 2.4% for *CLEF*-S, -M, and -L, respectively. Moreover, this made the performance of *CLEF* comparable to that of the supervised ECGFounder model, with differences in average AUROC of only 0.2%, −0.1%, and −0.3% for *CLEF*-S, -M, and -L, respectively, on lead I (the same lead ECGFounder is trained on too). Improvements are also observed when pretraining on lead II, of 3.4%, 2.0%, and 0.8% for *CLEF*-S, -M, and -L, respectively.

**Ablation on Handling Missing Metadata.** We assess the impact of our solution for handling missing metadata. As an ablation study, we report AUROC for all *CLEF* variants across 10 tasks including wearable datasets (MUSIC and Icentia11K) and single-lead ECG tasks (MIMIC-IV, PTB-XL, and Chapman) using lead I. Table 5 shows that without **M** in Eq. (4), the performance of all models degrades on the majority of tasks. On average, AUROC decreases by 1.9%, with a larger decrease on wearable datasets (4.0% drop) compared to lead I datasets (1.1% drop).

**Contribution of different loss components.** A study is conducted using CIFAR-100 to understand the contribution of different loss components. All models are initialized with ImageNet-1K pretrained ResNet-18 weights and further pretrained with: the standard contrastive loss $\mathcal{L}_{nce}$, the weighted contrastive loss $\mathcal{L}^w$, the dissimilarity alignment loss $\mathcal{L}^d$, $\mathcal{L}_{nce} + \mathcal{L}^d$, and our proposed $\mathcal{L}^w + \mathcal{L}^d$. A detailed experiment setup is provided in Appendix D.6. Our proposed $\mathcal{L}^w + \mathcal{L}^d$ achieves clearer separation with lower variance. Further details in Appendix D.6, Figure S2, and Table S15).

## 5  DISCUSSION AND CONCLUSION

We propose *CLEF*: clinically-guided contrastive learning to train single-lead ECG foundation models. On average, *CLEF* outperforms all baseline semi-supervised FMs across 18 clinical classification and regression tasks on 7 datasets. Furthermore, *CLEF* outperforms three self-supervised pretraining algorithms. When pretrained on the same lead as in downstream tasks, *CLEF* also performs comparably to ECGFounder, a state-of-the-art FM trained in a supervised manner on a labeled dataset. *CLEF* facilitates the creation of high-performance ECG FMs using only routinely-recorded metadata, without needing ECG-level annotations.

**Limitations.** We trained models on ECGs from a single hospital, which suit hospital-related tasks but are not representative of the general population. Performance on population-level tasks would likely improve with a more diverse cohort. The risk scores in this study are limited by only three out of seven SCORE2 input variables being available. Therefore, the scores cannot be considered a holistic assessment of cardiovascular risk, but instead are representative of real-world applications with incomplete metadata. Potentially richer embeddings could be obtained when using more complete metadata and, therefore, more precise risk scores. Finally, the single-lead ECGs used to assess performance in this study were measured using wet gel electrodes at the chest, and so are likely of higher quality than those typically measured by devices such as smartwatches and handheld ECGs.

**Future work.** One can investigate the utility of the proposed contrastive learning strategy for other modalities and health conditions. For instance, it may also be useful for developing photoplethysmography (PPG) FMs, since the PPG is also a cardiovascular signal and has been found to be associated with cardiovascular risk Weng et al. (2024). It could also be applied to other health conditions, such as incorporating the risk of deterioration in chronic respiratory conditions, in contrastive learning for respiratory signals. The practical application of this strategy would be aided by understanding which metadata variables contribute most to the quality of the representations produced by *CLEF*. A further ablation study, preferably conducted on a dataset with complete metadata, could identify the most important metadata variables and provide insight into how to apply this approach across diverse datasets with different metadata. Finally, we can go beyond traditional classification and regression tasks. Because the *CLEF*'s embedding space enables patient ECGs to be analyzed without prior metadata. It is possible to assess the proximity of current embeddings to prototypes of high-risk patients from the training set. This opens the door to clustering-based decision-making and other novel applications.

ETHICS STATEMENT

This study makes exclusive use of publicly available electrocardiogram datasets, each of which has independently undergone ethics approval by the respective data-providing institutions. All datasets comply with relevant privacy regulations, and no additional data collection was conducted. We acknowledge that biases may exist in the training data and recognize the importance of fair performance across diverse populations. Future evaluations on broader and more representative datasets will be essential for further improving the generalizability and fairness of our models. While our work demonstrates potential for advancing remote health monitoring and clinical decision support, we are mindful of the risks of misuse. Potential harmful applications could include unauthorized health surveillance, discriminatory practices in insurance or employment, unfair credit assessments, or the exploitation of sensitive health data for targeted marketing. We emphasize that this research is intended solely for beneficial healthcare and scientific purposes and strongly advocate for its responsible use. Our study adhered to established research ethics guidelines, and we declare no conflicts of interest. We further encourage interdisciplinary dialogue to anticipate and mitigate risks, develop appropriate governance frameworks, and safeguard individual rights. Recognizing the broader societal implications of AI in healthcare, we encourage ethical AI research and foster responsible deployment of these technologies.

REPRODUCIBILITY STATEMENT

We are committed to the reproducibility of our work. Models, code, and hyperparameters will be publicly released, accompanied by user-friendly tutorials to facilitate adoption and extension. Our work relies exclusively on publicly accessible datasets, which can be readily downloaded or requested from the respective data-hosting institutions. To further support reproducibility, we provide detailed descriptions of the datasets, model architectures, training and evaluation procedures, hyperparameters, and preprocessing methods used throughout the paper. Together, these resources will enable other researchers to replicate our results and build upon our contributions in future work.

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

## Supplementary Material
## CLEF: CLINICALLY-GUIDED CONTRASTIVE LEARNING FOR ELECTROCARDIOGRAM FOUNDATION MODELS

## A  Related Work

### A.1  ECG and ML

**Utility of single-lead ECGs compared to** 12**-lead.** Electrodes on the limbs and chest provide spatially diverse views of cardiac activity Kligfield et al. (2007), producing 12-lead ECGs to assess a range of heart conditions and abnormalities (Garcia and Holtz, 2001; Ribeiro et al., 2020). The growing usage of wearables and mobile health devices with single-lead ECG functionality (Hannun et al., 2019; Friedman, 2024; Attia et al., 2022) has opened new opportunities for continuous health monitoring beyond traditional care settings (Preejith et al., 2016; Majumder et al., 2018; Clark et al., 2018), with applications including early detection of cardiac events (Lin et al., 2019; Huda et al., 2020; Wang et al., 2022), long-term pattern tracking (Ravanshad et al., 2014; Majumder et al., 2018), and proactive health interventions (Melillo et al., 2015; Sopic et al., 2018; Bommasani et al., 2022).

Prior work Jiménez-Serrano et al. (2022) studied the Computing in Cardiology 2021 ECG dataset (covering 88,253 annotated 12-lead training recordings). They compared ML performance over 26 target cardiac conditions when processing only lead I versus all 12 leads. They found that 12 leads give the best overall performance, but using only lead I yields a modest average degradation rather than a collapse in accuracy: the average of G-metric (geometric mean of sensitivity and specificity) falls from 0.80 to 0.74. Specifically, among the 26 conditions, performance is similar (absolute G-metric change lower than 0.03) for 12 conditions; there is a moderate loss(G-metric drop from 0.04 to 0.07) for 7 conditions; and a significant drop (G-metric drop more than 0.07) for 7 conditions. On average, the drop in the G-metric is 0.06. Overall, this shows that single-lead (lead I) ECGs can reliably detect many rhythm-based abnormalities (making them attractive for wearables/screening), but they perform substantially worse for axis deviations, certain conduction blocks, and morphology-dependent findings that require precordial leads. Another observational study Angelaki et al. (2025) on 1,254 subjects showed that a single-lead ECG (lead I), when combined with other demographic features, can diagnose arterial hypertension with high accuracy (AUC 0.831, sensitivity 72%, specificity 82%), demonstrating that even single-lead ECGs contain sufficient diagnostic information when analyzed effectively. Additionally, a study by Ramirez et al. (2024) demonstrates that the standard 12-lead ECG contains redundant information when classifying cardiovascular diseases using CNNs. By selecting subsets of leads or applying transformations to the ECG signals, the authors evaluated how these adjustments influenced a CNN's diagnostic performance. Their main finding is that carefully optimizing input configurations, hence reducing redundancy while preserving essential information, can improve deep learning model performance and enable efficient diagnostics even with fewer leads.

**Diagnostic potential and limitations of machine learning for ECG analysis.** A proof-of-concept study by Sun et al. (2022) demonstrates that a ResNet model trained on large-scale, population-based ECG datasets can accurately predict a wide range of diseases, including many non-cardiovascular conditions. Using over 1.5 million ECGs linked to 11K unique WHO ICD codes from 240K patients across 26 hospitals in Canada, the authors identified 700 disease categories with sufficient data for modeling. Models achieved strong discriminative performance (AUROC > 80%) for 80 disease categories, with 18 categories exceeding AUROC > 90% (including non-cardivascular conditions such as silicosis, type 1 diabetes mellitus, liver diseases, behavioral disorder due to drug use, and some maternal diseases during pregnancy). Despite excellent AUROC values, precision was limited for many conditions due to their low prevalence, suggesting greater utility for rule-out screening rather than definitive diagnosis. The findings highlight the untapped diagnostic potential of ECGs for diverse diseases, while also noting that predictions may reflect correlated comorbidities or patient

Table S1: Available metadata for each database. M-iv-ecg is short for MIMIC-IV-ECG, G12EC is short for Georgia 12-lead ECG Challenge, and Phy2021 is short for PhysioNet 2021.

| Dataset | M-iv-ecg (2023) | CPSC2018 (2018) | Chapman (2020) | PTB-XL (2020) | CODE-15 (2021) | G12EC (2020) | HEEDB (2025) | Phy2021 (2021) |
|---|---|---|---|---|---|---|---|---|
| Lead number | 12 | 12 | 12 | 12 | 12 | 12 | 12 | 12 |
| Record number | 800,035 | - | 45,152 | 21,799 | 345,779 | - | 10,471,531 | - |
| Patient number | 161,352 | 50,165 | 45,152 | 18,869 | 233,770 | - | 1,818,247 | - |
| Sample rate (Hz) | 500 | 500 | 500 | 500 | 400 | 500 | 500 | varied |
| Duration | 10 s | 10 s | 10 s | 10 s | 10 s | varied | 10 s | varied |
| Sex | ✔ | ✔ | ✔ | ✔ | ✔ | ✔ | ✔ | ✔ |
| Sex at Birth | | | | | | | ✔ | |
| Gender identity | | | | | | | | ✔ |
| Age | ✔ | ✔ | ✔ | ✔ | ✔ | ✔ | ✔ | ✔ |
| Weight | ✔ | | | ✔ | | | | |
| Height | ✔ | | | ✔ | | | | |
| BMI | ✔ | | | | | | | |
| Race | ✔ | | | | | | ✔ | |
| Ethnicity | | | | | | | ✔ | |
| Marital status | | | | | | | ✔ | |
| Religion | | | | | | | ✔ | |
| Language | | | | | | | ✔ | |
| Veteran | | | | | | | ✔ | |
| Education | | | | | | | ✔ | |
| Date of Birth | | | | | | | ✔ | |
| Date of Death | ✔ | | | | | | ✔ | |
| Last visit date | | | | | | | ✔ | |
| SNOMED | | | ✔ | | | | | |
| sinus rhythm | | | | | ✔ | | | |
| AF | | | | | ✔ | | | |
| bundle branch block | | | | | ✔ | | | |
| Used Studies | Tian et al. Liu et al. Jin et al. McKeen et al. | Mehari and Strodthoff Wang et al. | Na et al. | Mehari and Strodthoff Wang et al. | Na et al. | Wang et al. | Li et al. | McKeen et al. |

characteristics rather than direct disease-specific ECG changes, and that further targeted, clinically adjudicated studies are needed before deployment. Another study Kim et al. (2024) using data from 919 patients found that time-dependent follow-up features contributed more strongly to predicting heart failure rehospitalization than admission or discharge variables, highlighting the dynamic nature of heart failure risk and underscoring the importance of ongoing monitoring and medication adherence during the post-discharge period for more accurate risk stratification and targeted intervention.

## A.2 CONTRASTIVE APPROACHES

**Adapting self-supervised learning for ECGs.** Classic self-supervised methods based on instance discrimination and latent forecasting are adapted by Mehari and Strodthoff (2022) to the ECG domain. They utilize time-series-specific data augmentations (including physiological noise models), modifying the CPC architecture to suit ECG's temporal resolution, employing a joint 12-lead encoder, and systematically evaluating multiple frameworks on large public datasets for their impact on downstream performance, label efficiency, and robustness. They find out that contrastive predictive coding (CPC) with these adaptations achieves near-supervised linear evaluation performance and significantly improves downstream accuracy, data efficiency, and robustness to physiological noise.

**Multi-objective contrastive learning.** A combination of two contrastive losses for pretraining is used by Oh et al. (2022), where a Wav2Vec architecture captures local lead-level temporal features and a contrastive multi-segment coding architecture from Kiyasseh et al. (2021) captures global patient-level context. The total loss is the sum of both, enabling the model to learn fine-grained intra-signal information and broader inter-patient relationships. Also, during pretraining, the authors introduce Random Lead Masking, where each ECG lead is randomly masked with a fixed probability to simulate reduced-lead scenarios. This augmentation makes the single pretrained model robust to arbitrary lead configurations during fine-tuning.

## A.3 ECG DATASET OVERVIEW

The ECG datasets used in prior work exhibit substantial variability in scale, metadata richness, and clinical scope, directly impacting foundation model development capabilities. A detailed

Table S2: ECG diagnostic labels in the PTB-XL dataset. The dataset contains 42 diagnostic labels in a 3-level hierarchy: superclass, subclass, and specific diagnosis. The **Label** column denotes the specific diagnostic label, while the **Subclass** and **Superclass** column indicates its corresponding subclass and superclass categories.

| Label | Description | Subclass | Superclass |
|---|---|---|---|
| *Conduction Disturbances (CD)* | | | |
| LAFB | Left anterior fascicular block | LAFB/LPFB | CD |
| LPFB | Left posterior fascicular block | LAFB/LPFB | CD |
| IRBBB | Incomplete right bundle branch block | IRBBB | CD |
| ILBBB | Incomplete left bundle branch block | ILBBB | CD |
| CRBBB | Complete right bundle branch block | CRBBB | CD |
| CLBBB | Complete left bundle branch block | CLBBB | CD |
| AVB | First degree AV block | AVB | CD |
| 3AVB | Third degree AV block | AVB | CD |
| 2AVB | Second degree AV block | AVB | CD |
| IVCD | Non-specific intraventricular conduction disturbance | IVCD | CD |
| WPW | Wolff-Parkinson-White syndrome | WPW | CD |
| *Hypertrophy (HYP)* | | | |
| LVH | Left ventricular hypertrophy | LVH | HYP |
| LAO/LAE | Left atrial overload/enlargement | LAO/LAE | HYP |
| RVH | Right ventricular hypertrophy | RVH | HYP |
| RAO/RAE | Right atrial overload/enlargement | RAO/RAE | HYP |
| SEHYP | Septal hypertrophy | SEHYP | HYP |
| *Myocardial Infarction (MI)* | | | |
| IMI | Inferior myocardial infarction | IMI | MI |
| ILMI | Inferolateral myocardial infarction | IMI | MI |
| IPLMI | Inferoposterolateral myocardial infarction | IMI | MI |
| IPMI | Inferoposterior myocardial infarction | IMI | MI |
| INJIN | Subendocardial injury in inferior leads | IMI | MI |
| INJIL | Subendocardial injury in inferolateral leads | IMI | MI |
| ASMI | Anteroseptal myocardial infarction | AMI | MI |
| AMI | Anterior myocardial infarction | AMI | MI |
| ALMI | Anterolateral myocardial infarction | AMI | MI |
| INJAS | Subendocardial injury in anteroseptal leads | AMI | MI |
| INJAL | Subendocardial injury in anterolateral leads | AMI | MI |
| INJLA | Subendocardial injury in lateral leads | AMI | MI |
| LMI | Lateral myocardial infarction | LMI | MI |
| PMI | Posterior myocardial infarction | PMI | MI |
| *ST/T Changes (STTC)* | | | |
| NDT | Non-diagnostic T abnormalities | STTC | STTC |
| DIG | Digitalis effect | STTC | STTC |
| LNGQT | Long QT interval | STTC | STTC |
| ANEUR | ST-T changes compatible with ventricular aneurysm | STTC | STTC |
| EL | Electrolytic disturbance or drug effect | STTC | STTC |
| NST | Non-specific ST changes | NST | STTC |
| ISC | Non-specific ischemic changes | ISC | STTC |
| ISCIN | Ischemic changes in inferior leads | ISCI | STTC |
| ISCIL | Ischemic changes in inferolateral leads | ISCI | STTC |
| ISCAL | Ischemic changes in anterolateral leads | ISCA | STTC |
| ISCAS | Ischemic changes in anteroseptal leads | ISCA | STTC |
| ISCLA | Ischemic changes in lateral leads | ISCA | STTC |
| ISCAN | Ischemic changes in anterior leads | ISCA | STTC |
| *Normal (NORM)* | | | |
| NORM | Normal ECG | NORM | NORM |

summarization of the current available ECG dataset is tabulated in Table S1. Among the datasets, MIMIC-IV-ECG emerges as a particularly well-suited one for health risk assessment applications due to its combination of large-scale clinical data (161,352 patients) and a relatively inclusive metadata,

Table S3: Form labels and label descriptions in the PTB-XL dataset. Labels represent morphological and rhythm abnormalities detected in ECG recordings.

| Label | Description | Label | Description |
|-------|-------------|-------|-------------|
| ABQRS | Abnormal QRS | LOWT | Low amplitude T-waves |
| DIG | Digitalis effect | LPR | Prolonged PR interval |
| HVOLT | High QRS voltage | LVOLT | Low QRS voltages |
| INVT | Inverted T-waves | NDT | Non-diagnostic T abnormalities |
| LNGQT | Long QT interval | NST | Non-specific ST changes |
| NT | Non-specific T-wave changes | PAC | Atrial premature complex |
| PRC(S) | Premature complex(es) | PVC | Ventricular premature complex |
| QWAVE | Q waves present | STD | Non-specific ST depression |
| STE | Non-specific ST elevation | TAB | T-wave abnormality |
| VCLVH | Voltage criteria for LVH | | |

Table S4: Rhythm labels and label descriptions in the PTB-XL dataset. Labels represent various cardiac rhythm patterns and arrhythmias detected in ECG recordings.

| Label | Description | Label | Description |
|-------|-------------|-------|-------------|
| AFIB | Atrial fibrillation | PACE | Normal functioning artificial pacemaker |
| AFLT | Atrial flutter | PSVT | Paroxysmal supraventricular tachycardia |
| BIGU | Bigeminal pattern (unknown origin) | SARRH | Sinus arrhythmia |
| SBRAD | Sinus bradycardia | SR | Sinus rhythm |
| STACH | Sinus tachycardia | SVARR | Supraventricular arrhythmia |
| SVTAC | Supraventricular tachycardia | TRIGU | Trigeminal pattern (unknown origin) |

including gender, age, and SBP. In contrast, alternative datasets present significant limitations: smaller-scale collections like Chapman and PTB-XL lack the statistical power for robust foundation model training, while metadata-sparse datasets preclude comprehensive health risk modeling. Although HEEDB offers a superior scale (1.8 million patients), its restricted accessibility limits reproducibility and benchmarking capabilities.

## B  DETAILS OF THE DATASETS USED FOR PRETRAINING AND DOWNSTREAM EVALUATION

**MIMIC-IV-ECG** dataset is a large-scale clinical ECG database containing 800,035 12-lead diagnostic recordings collected from 161,352 unique patients.[4] Each ECG is 10 seconds in length and sampled at 500Hz. Corresponding metadata can be matched to each patient using the unique patient code.[5] A more detailed procedure of aligning metadata for each ECG sample is provided in Appendix B.1. We use this dataset in 2 phases: (1) for pretraining, we use the first unique ECG recording from each patient and train over all distinct patients, and (2) for downstream evaluation, we follow prior work (Li et al., 2024) who obtained the labels for the LVEF estimation task from the discharge section of MIMIC-IV-Notes. Specifically, LVEF values are provided as continuous labels for the regression task, while an LVEF of 50% or higher is considered normal and below 50% abnormal, thereby defining a binary classification task. We sequentially split the dataset into training (60,329 samples, 80%), validation (7,541 samples, 10%), and test sets (7,542 samples, 10%).

**MUSIC** dataset (MUerte Subita en Insuficiencia Cardiaca) focuses on assessing cardiac mortality and sudden cardiac death (SCD) in ambulatory patients with chronic heart failure (CHF).[6] It contains 992 patients with CHF consecutively enrolled from the specialized HF clinics of eight University Spanish Hospitals between April 2003 and December 2004. All patients are measured with a 3-lead resting electrocardiogram (ECG), or a 3-lead Holter ECG. In this study, we focus on using the first lead of the collected Holter ECG signal. In the original dataset, outcome labels (non-cardiovascular death, sudden cardiac death, or pump-failure death) are provided at the patient level. For each patient, we extract the first 10 seconds of ECG recordings with non-zero signals. For the downstream

---

[4]MIMIC-IV-ECG dataset available at: `physionet.org/content/mimic-iv-ecg`.

[5]Metadata for MIMIC-IV available at: `physionet.org/content/mimiciv`.

[6]MUSIC dataset available at: `physionet.org/content/music-sudden-cardiac-death`.

Table S5: Beat Symbol Definitions with Icentia11K data set

| Symbol | Beat Description |
|--------|------------------|
| N | Normal |
| S | ESSV (PAC): Premature or ectopic supraventricular beat, premature atrial contraction |
| V | ESV (PVC): Premature ventricular contraction, premature ventricular contraction |

Table S6: Rhythm symbol definitions with Icentia11K data set.

| Symbol | Rhythm Type | Rhythm description |
|--------|-------------|--------------------|
| (N . . . ) | NSR (Normal sinus rhythm) | - |
| (AFIB . . . ) | AFib (Atrial fibrillation) | Irregular rhythm with absent P waves and irregular RR intervals |
| (AFL . . . ) | AFlutter (Atrial flutter) | Regular atrial arrhythmia with sawtooth flutter waves |

experiments, we take 188 samples (20%) as a held-out test set, while the remaining 655 samples are for training and 93 for validation.

**PTB-XL** dataset is a large-scale ECG dataset that has been widely used in prior research (cite) for evaluating model capacity in signal pattern and disease identification.[7] We adopt the preprocessing and label alignment procedures described in the original dataset publication.[8] Details of the tasks of PTB-XL are tabulated in Tables S2, S3, and S4 To ensure fair comparison, we follow prior work (Wagner et al., 2020) and adopt a data partitioning of $[80/10/10]\%$, yielding 16,832, 2,100, and 2,098 samples for training, validation, and testing phases, respectively.

**Icentia11K** dataset is a wearable dataset that contains ECG signals collected from single-lead chest-mounted wearable devices.[9] The sample rate for the dataset is 250 Hz. Beat and rhythm labels are extracted from the dataset's annotation files, focusing on three beat types (Normal, Supraventricular, Ventricular) and three rhythm types (Normal Sinus Rhythm, Atrial Fibrillation, Atrial Flutter), see Tables S5 and S6 for a more detailed description. Note that the dataset also includes an undefined beat class, which we omit here as it does not correspond to a physiologically interpretable beat type. For both tasks, segments are generated by slicing 10-second windows starting from the annotated event and assigning numerical labels for classification. To ensure balanced representation across patients, we employ a patient-stratified sampling strategy: for each patient, we search their recordings for each target beat type and randomly select one representative instance, continuing until all types are found or available segments are exhausted. This results in relatively balanced distributions across beat classes (N: 10,866; S: 9,844; V: 9,287), whereas rhythm classes remain imbalanced due to the limited prevalence of atrial fibrillation and atrial flutter (N: 10,239; AFib: 743; AFL: 516). For experimental evaluation, the dataset is partitioned into training, validation, and test sets (90%/10%/10%), comprising $20,994/4,532/4,511$ samples for beat classification and $8,042/1,723/1,733$ samples for rhythm classification.

**Chapman** dataset is a collection of 12-lead ECG recordings from 45,152 patients labeled by clinical experts to support research in arrhythmia and cardiovascular disease detection.[10] The dataset contains 10-second recordings sampled at 500 Hz, featuring 11 common cardiac rhythms and 67 additional cardiovascular conditions, all validated through a rigorous multi-physician review process with senior physician arbitration for diagnostic disagreements. We follow prior work (Jin et al., 2025) and use a refined version of the dataset, which contains 23,026 ECG recordings with 38 distinct labels. The samples are then split into training ($16,546$ samples, 70%), validation ($1,860$ samples, 10%), and test sets ($4,620$ samples, 20%).

**MC-MED** dataset is a multimodal collection of emergency department visits from 118,385 adult patients at Stanford Health Care between 2020 and 2022.[11] The dataset combines continuous physiological monitoring (lead II ECG, photoplethysmography, respiration waveforms), clinical data

---

[7]PTB-XL dataset available at: `physionet.org/content/ptb-xl`.

[8]Data preprocessing code for PTB-XL available at: `github.com/helme/ecg_ptbxl_benchmarking`.

[9]Icentia11K dataset available at: `physionet.org/content/icentia11k-continuous-ecg`.

[10]Chapman dataset available at: `physionet.org/content/ecg-arrhythmia`.

[11]MC-MED dataset available at: `physionet.org/content/mc-med`.

(demographics, medical histories, laboratory results, medications, radiology reports), and temporal visit outcomes. It covers emergency department patients during and after the COVID-19 pandemic with granular physiological measurements.

The prediction targets encompass both classification and regression tasks across the clinical care continuum. For classification, there are 3 tasks: (1) emergency department (ED) disposition, which determines patient placement after ED assessment. This includes 4 classes: Discharge (outpatient), Inpatient (hospital admission), Observation (extended monitoring without full admission), and ICU (critical care); (2) Discharge (DC) disposition, classification of patient outcomes at hospital discharge. We categorized the outcome into 5 categories: home care, care facility, hospital transfer, psychiatric care, and death. and (3) Triage acuity assessment, which is predicting clinical urgency as determined by healthcare professionals during initial patient evaluation, including 5 tasks: Resuscitation, emergent, urgent, Semi-Urgent, and Non-Urgent. The regression tasks involve continuous prediction of systolic blood pressure (SBP) and diastolic blood pressure (DBP) values. We follow the chronological splitting provided in the original dataset based on patient admission dates, and allocate the first $78\%$ (37,438 samples) for training, the following $11\%$ (5,540 samples) for validation, and the most recent $11\%$ (5,572 samples) for testing. Note that the exact number may differ from the original dataset, as we only include patients with lead II ECG recordings.

**Aurora BP** dataset is a collection of simultaneous multi-modal physiological recordings collected from 1,221 diverse participants,[12] serving the purpose for cuffless blood pressure research. The dataset contains synchronized tonometry, photoplethysmography (PPG), electrocardiography (ECG), accelerometry, and reference blood pressure measurements collected during both laboratory and 24-hour ambulatory monitoring phases, with participants spanning a wide range of ages and hypertensive status to ensure real-world applicability. The prediction targets are systolic blood pressure (SBP) and diastolic blood pressure (DBP), with the task formulated as the estimation of their actual values through regression. We use a $70\%/15\%/15\%$ train/validation/test split on patient ID, resulting in 786/169/169 patients, which corresponds to 9,237/1,913/1,854 samples for training, validation, and testing, respectively.

### B.1 METADATA ALIGNMENT WITH MIMIC-IV-ECG DATASET

The MIMIC-IV-ECG database provides raw electrocardiogram (ECG) waveforms together with patient identifiers and relative timestamps, but does not directly include demographic or clinical metadata. To match each ECG with patient information, we aligned it with the corresponding records in MIMIC-IV,[13] which contains both static demographics (e.g., sex, age at admission) and time-stamped clinical observations (e.g., blood pressure). Each ECG was first linked to the corresponding patient using the unique patient identifier, which is shared across MIMIC-IV and MIMIC-IV-ECG. Demographic variables that remain constant over time were directly assigned to all ECGs of the same patient. For time-varying clinical measurements, we selected the most recent value recorded at or before the ECG timestamp, ensuring that the metadata reflected the patient's state at the time of acquisition. If no prior measurement was available, the variable was marked as missing, and no forward-filling across admissions or imputation beyond this step was performed. This procedure ensures that every ECG recording is annotated with the most temporally relevant metadata.

## C SUPPLEMENTARY EXPERIMENT SETUP AND BASELINES

### C.1 EXPERIMENT SETUP FOR DOWNSTREAM EVALUATION

This section supplements the experiment setup in section 3 of the main paper. All experiments were conducted using a Linux server (Ubuntu 22.04.5) with 4 NVIDIA L40S GPUs, 2 Intel Xeon Gold 6542Y CPUs. Code will be made available.

**Pretraining:** All experiments are conducted under fixed hyperparameter settings. All models are pretrained for 100 epochs with a batch size of 64 using the AdamW optimizer, except for the experiments for pretraining on a specific lead due to a reduced data diversity (section 4, pretraining

---

[12]A sample data of Aurora BP is available at: github.com/microsoft/aurorabp-sample-data. Access can be provided via application.

[13]MIMIC-IV database available at: physionet.org/content/mimiciv/3.1/

on a specific lead), where we pretrain for 10 epochs. A learning rate of $1 \times 10^{-4}$ and a weight decay of $5 \times 10^{-5}$ are employed, with a Cosine Annealing Learning Rate scheduler to adjust the learning rate. Training stops if the validation loss does not decrease for 20 consecutive epochs. Seed is fixed to 42 for all pretraining. For parameters specific to contrastive pretraining, we set the clinical-guidance coefficient to $\alpha = 0.2$ and temperature to $\tau = 0.07$, consistent with prior literature (Wu et al., 2018; He et al., 2020). The effect of $\alpha$ and $\tau$ is further examined in Appendix D.8, Table S17, and Table S18.

**Downstream tasks:** For downstream evaluation, training is also conducted for 100 epochs with a batch size of 64. We use binary cross-entropy loss for classification and L1 loss for regression tasks. Model parameters are optimized using Adam with a learning rate of $1 \times 10^{-3}$ and a weight decay of $1 \times 10^{-5}$. The learning rate is adjusted using a Cosine Annealing with Warm Restarts scheduler. Training stops if the validation loss does not decrease for 5 consecutive epochs. Seed is fixed to 42 for all downstream evaluation, except for the seed robustness evaluation in Appendix D.1.

**Downstream preprocessing:** For all downstream tasks, the ECG signals are resampled to 500 Hz. For datasets containing recordings longer than 10 seconds, we keep the first 10 seconds as our input. We preprocessed all ECG signals using a 5th-order Butterworth bandpass filter with cutoff frequencies of 0.67 Hz and 40 Hz to remove low-frequency baseline wander (e.g. due to breathing) whilst maintaining heart rate frequencies, and to remove high-frequency noise (e.g. power-line interference) whilst maintaining fundamental frequencies of P, QRS, and T-waves. The input ECG samples are then standardized using z-scoring, an operation that transforms the data to zero mean and unit standard deviation.

**Regression specific setup:** In all regression tasks, the target labels are also z-scored using statistics computed from the training set. Predictions are de-normalized back to the original scale before comparison with the ground-truth labels. The errors are evaluated on the original scale.

## C.2 DATA AUGMENTATION FOR CONTRASTIVE PRETRAINING

This section supplements the **stochastic data augmentation** in section 3.

**Simulate physiological noise** We inject physiological noise collected from the MIT-BIH noise stress test database (Sološenko et al., 2021a), as a form of data augmentation that encourages the model to learn representations invariant to electrode movement and motion-induced artifacts, etc.[14] The noise injection is achieved by selecting the noise signal from the corresponding lead, given by $\hat{\mathbf{e}}_s^l = \mathbf{e}_s^l + \phi \times \mathbf{n}^l$, where $\phi$ controls the noise intensity, $\mathbf{n}^l$ is the noise vector of the $l$-th lead. We provide visualization in Figure S1 of how different noises influence and augment the original signal. Note that we set $\phi = 0.02$ following prior work (Goldberger et al., 2000; Sološenko et al., 2021b), a parameter also used in the main experiment.

---

[14]The noise is available at: `physionet.org/content/ecg-ppg-simulator-arrhythmia`.

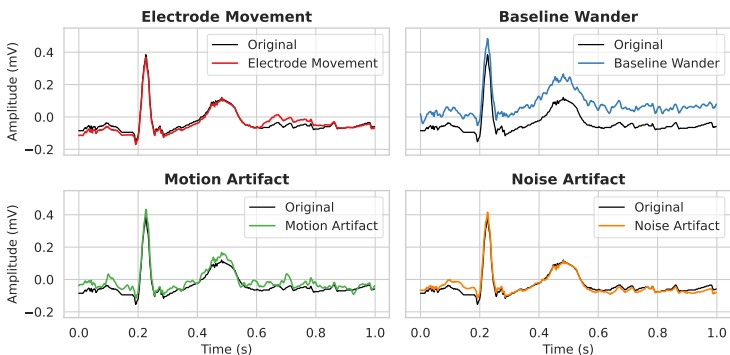

Figure S1: Physiological noise for single-lead ECG contrastive learning. We include four common sources of signal degradation in wearable ECG devices: electrode movement artifacts, baseline wander, motion-induced distortions, and additive noise. Original signals are shown in black, with augmented versions in color.

Table S7: ResNeXt1D Model Architecture Configurations for *CLEF* variants. **Left:** Shared hyperparameters of all three variants. **Right:** Detailed stage-wise model configurations.

| Parameter | CLEF-S | CLEF-M | CLEF-L |
|---|---|---|---|
| **Model Size** | 448K | 30.7M | 296M |
| **Hidden Dimension** $d$ | 32 | 64 | 128 |
| **Ratio** $\iota$ | 0.5 | 1.0 | 1.5 |
| **Groups Width** $g$ | 8 | 16 | 32 |
| **Number of Stages** $\varpi$ | 6 | 7 | 9 |

| Stage | CLEF-S $h / \gamma$ | CLEF-M $h / \gamma$ | CLEF-L $h / \gamma$ |
|---|---|---|---|
| 1 | 32 / 1 | 64 / 2 | 128 / 2 |
| 2 | 64 / 1 | 160 / 2 | 256 / 3 |
| 3 | 64 / 2 | 160 / 2 | 256 / 3 |
| 4 | 128 / 2 | 400 / 3 | 512 / 4 |
| 5 | 128 / 2 | 400 / 3 | 512 / 4 |
| 6 | 256 / 2 | 1024 / 4 | 1024 / 5 |
| 7 | – | 1024 / 4 | 1024 / 5 |
| 8 | – | – | 2048 / 6 |
| 9 | – | – | 2048 / 6 |

**Random mask**   Randomly masking segments of the ECG signal encourages the model to learn representations that are invariant to missing information. In our experiments, we set the masking probability to p = 0.2, with each selected segment having 10% of its signal randomly masked.

## C.3   BACKBONE RESNET MODELS

*CLEF* is built upon a multi-layer ResNeXt1D architecture, where each stage contains multiple residual blocks. We implement three variants, i.e. small, medium, and large, for our proposed *CLEF* model with different computational complexities. All variants follow the same architectural principles but differ in depth, width, and parameter count.

**Convolution layer.** The network begins with an initial convolutional layer with 1 channel, a kernel size of 16, and a stride of 2. This layer takes the input ECG signal $\mathbf{e}_s^l \in \mathbb{R}^t$ from the $s$-th sample and $l$-th lead, where t denotes the signal length. that takes input ECG signal $\mathbf{e}_s^l \in \mathbb{R}^t$ of $s$-th sample and $l$-th lead, where $t$ denotes the signal length. The input $\mathbf{e}_s^l$ is first transformed by the layer and passed through a swish activation, yielding the hidden state $\mathbf{H}_s^l \in \mathbb{D}^{d \times \ell}$, where $\ell = \frac{t}{2}$ is the sequence length, and the hidden dimension $d$ is set to 32, 64, or 128 for small, medium, and large variants, respectively. For brevity, we omit the sample & lead indices $s$ and $l$ in the intermediate representations.

**Residual stages.** The network then processes $\mathbf{H}_s^l$ through a sequence of $\varpi$ residual stages. Each stage consists of $\gamma$ residual blocks. Each residual block processes features through three convolution layers sequentially:

(1) A $1 \times 1$ convolution layer with stride 1 transforms $\mathbf{H}_s^l$ to an intermediate dimension $\mathbf{H}^{(1)} \in \mathbb{R}^{d^{(1)} \times \ell}$, where $d^{(1)} = d \times \iota$, with $\iota \in \{0.5, 1.0, 1.5\}$ is a predefined ratio parameter for small, medium, and large models, respectively.

(2) A $16 \times 16$ convolution layer with stride 1 is applied to $\mathbf{H}^{(1)}$, producing the layer output $\mathbf{H}^{(2)} \in \mathbb{R}^{d^{(1)} \times \ell}$. Specifically, the input is first divided into $d^{(1)}/g$ groups along channel dimension, where $g \in \{8, 16, 32\}$ specifies the group width for the small, medium, and large variants, respectively. Each group is convolved independently with its own set of filters, and the resulting features are concatenated along the channel dimension to form the output $\mathbf{H}^{(2)}$.

(3) Finally, a $1 \times 1$ convolution layer maps $\mathbf{H}^{(2)}$ to the output tensor $\mathbf{H}^{(3)} \in \mathbb{R}^{h \times \ell}$, where $h$ denotes the final output channel dimension of the residual block.

A swish activation is used between layers. Following the final convolutional output of the last block of each stage, the output matrix $\mathbf{H}^{(3)}$ is first averaged along sequence dimension, forming vector $\mathbf{h}_s \in \mathbb{R}^h$. This vector $\mathbf{h}_s$ is passed through a swish-activated 2-layer perception (hidden dimension is $\frac{h}{2}$, output dimension is $h$), followed by a sigmoid activation, forming channel-wise gating scores $\mathbf{h}_r \in \mathbb{R}^h$. Each channel vector of $\mathbf{H}^{(3)}$ is multiplied by the corresponding element of $\mathbf{h}_r$, and the reweighted channel vectors collectively form the residual tensor $\mathbf{H}_r \in \mathbb{R}^{h \times \ell}$. This residual term is then added with $\mathbf{H}^{(3)}$ to form the output of the state, given by $\mathbf{H}_o = \mathbf{H}^{(3)} + \mathbf{H}_r$. The final output of the last stage is averaged over the sequence dimension to obtain the feature embedding of the input signal, denoted as $\mathbf{z} \in \mathbb{R}^h$. Table S7 details the component parameters of each variant:

Table S8: SCORE2 model coefficients ($\beta_i$) by gender and age group ($<70$ vs. $\geq 70$ years).

| Coefficient | Male $< 70$ | Female $< 70$ | Male $\geq 70$ | Female $\geq 70$ |
|:---:|:---:|:---:|:---:|:---:|
| $\beta_1$ | 0.3742 | 0.4648 | 0.0634 | 0.0789 |
| $\beta_2$ | 0.6012 | 0.7744 | 0.3524 | 0.4921 |
| $\beta_3$ | 0.2777 | 0.3131 | 0.0094 | 0.0102 |
| $\beta_4$ | 0.6457 | 0.8096 | 0.4245 | 0.6010 |
| $\beta_5$ | 0.1458 | 0.1002 | 0.0850 | 0.0605 |
| $\beta_6$ | -0.2698 | -0.2606 | -0.3564 | -0.3040 |
| $\beta_7$ | -0.0755 | -0.1088 | -0.0247 | -0.0255 |
| $\beta_8$ | -0.0255 | -0.0277 | -0.0005 | -0.0004 |
| $\beta_9$ | -0.0281 | -0.0226 | 0.0073 | -0.0009 |
| $\beta_{10}$ | 0.0426 | 0.0613 | 0.0091 | 0.0154 |
| $\beta_{11}$ | -0.0983 | -0.1272 | -0.0174 | -0.0107 |
| $S_0(t)$ | 0.9605 | 0.9776 | 0.7576 | 0.8082 |
| $c$ | 0 | 0 | 0.0929 | 0.2290 |

## C.4 BASELINE MODELS

**ST-MEM** (Na et al., 2024) is a self-supervised learning framework specifically designed to capture spatio-temporal dependencies in 12-lead ECG signals. It employs a spatio-temporal masked auto-encoder that reconstructs randomly masked signal segments, enabling the model to learn rich ECG representations.

**KED** (Tian et al., 2024) is an ECG foundation model pretrained with raw ECG signal and textual input (ECG reports). They used contrastive learning to align the representation of two modalities, enhancing diagnostic performance.

**ECGFounder** (Li et al., 2024) is a foundation model for ECG trained in a supervised manner on large-scale labeled datasets for multi-label classification. To address the challenge of missing labels that are commonly present in long-tailed clinical datasets, the authors propose a modified loss function that enables more robust and balanced representation learning despite label sparsity.

**Moirai** (Woo et al., 2024) is a foundation model for time series designed for general-purpose forecasting. It supports both univariate and multivariate forecasting. For multivariable input, to get an aligned sequence, they reshape the input sequence into a single, aligned sequence that integrates both temporal and variable dimensions, enabling unified processing through the transformer architecture.

**Moment** (Goswami et al., 2024) is a transformer-based model built on the T5 architecture, designed for univariate time series tasks. It supports both sequence-to-sequence forecasting and time series classification, offering a unified and scalable foundational framework for temporal modeling.

## C.5 BASELINE PRETRAINING METHODS

**SimCLR** (Chen et al., 2020) is a contrastive learning framework whose objective is to bring the representations of augmented views of the same sample closer together, while pushing apart representations of different samples. It relies on a large set of diverse negative samples, which are typically achieved through large batch sizes, to provide effective contrastive learning signals.

**BYOL** (Grill et al., 2020) is a self-supervised learning method that does not require negative samples. It proposed to have an online network and a target network. The online network is trained to predict the target network's representation of the same augmented input. The target network is updated via an exponential moving average of the online parameters.

**MoCo** (He et al., 2020) is also a modified contrastive learning framework. Instead of relying on negatives sampled directly from the dataset, they propose to have a dynamic memory bank that stores encoded representations from previous batches. This memory bank is updated using a momentum encoder, which ensures stable and consistent representations over time. The stored representations serve as negative samples for contrastive learning, providing a large and diverse set of negatives without requiring large batch sizes.

## C.6 DETAILS IN OBTAINING SCORE2 RISK SCORES

The SCORE2 risk score estimates the 10-year risk of fatal and non-fatal cardiovascular disease.[15] Demographic information (e.g. sex and age) and clinical measurements (e.g. blood pressure) are used to predict cardiovascular risk (working group and risk collaboration, 2021a;b). Specifically, for sample $s$, the score uses 7 features, including: age ($a_s^1$, in years), gender ($a_s^2 \in \{\text{male}, \text{female}\}$), smoking status ($a_s^3 \in \{0, 1\}$), systolic blood pressure (SBP, $a_s^4$ in mmHg), diabetes status ($a_s^5 \in \{0, 1\}$), total cholesterol ($a_s^6$ in mmol/L), and high-density lipoprotein (HDL) cholesterol ($a_s^7$ in mmol/L).

**Detailed risk score computation.** The 7 covariates are first scaled by subtracting their corresponding mean values and then scaling by their pre-defined units (e.g., 5 years for age, 20 mmHg for SBP), except for the 3 binary features, gender, smoking status, and diabetes status. Given sample $s$, the transformed variables are given as:

$$u_s^1 := \frac{a_s^1 - 60}{5}, \quad u_s^2 := a_s^3, \quad u_s^3 := \frac{a_s^4 - 120}{20}, \quad u_s^4 := a_s^5, \quad u_s^5 := a_s^6 - 6, \quad u_s^6 := \frac{a_s^7 - 1.3}{0.5} \ . \tag{S1}$$

These normalized covariates, collected in vector $\mathbf{u} = [u_s^1, u_s^2, \cdots, u_s^6] \in \mathbb{R}^6$, are then transformed with parameters $\mathbf{b}_1 = [\beta_1, \beta_2, \beta_3, \beta_4, \beta_5, \beta_6] \in \mathbb{R}^6$, and age-dependent transformation $\mathbf{b}_2 = [0, \beta_7, \beta_8, \beta_9, \beta_{10}, \beta_{11}] \in \mathbb{R}^6$, given by:

$$\chi_s = \mathbf{b}_1 \mathbf{u}_s^\top + u_1 \mathbf{b}_2 \mathbf{u}_s^\top \ , \tag{S2}$$

where $\chi$ marks an intermediate score. Different parameter sets $\mathbf{b}_1$ and $\mathbf{b}_2$ are used for males and females, and also for different age groups. Details are tabulated in Table S8.

The health risk score $r$ of a patient sample is then obtained using an exponential transformation of $\chi$:

$$r_s = 1 - S_0(t)^{\exp(\chi_s - c)} \ , \tag{S3}$$

where $S_0(t)$ is the baseline survival function at time $t$ (10 years) and $c$ is an offset. Different $S_0(t)$ and $c$ are also used for different gender and age groups. Details are tabulated in Table S8. Note that the original SCORE2 calculation includes a step for regional calibration. As regional information is unavailable in our dataset, this step was omitted. We directly used the uncalibrated 10-year risk score, with no further region-specific adjustment applied.

The SCORE2 risk score initially uses 7 variables, but because MIMIC-IV-ECG only records the age, gender, and systolic BP, we infer missing values with simple imputation strategies. Smoking and diabetes status were assumed to be absent if not recorded (i.e., set to 0). The missing total cholesterol and HDL cholesterol values were estimated using population-based reference values stratified by sex, with added Gaussian noise to reflect natural biological variation. Specifically, the total cholesterol was imputed as $5.2 \pm 0.5$ mmol/L and HDL cholesterol as $1.3 \pm 0.2$ mmol/L, based on the mean values of age group 40-45 (working group and risk collaboration, 2021a). Additionally, there are 15 records with missing age, which we impute as $40$. The number of missing values is accounted for in the **handling missing metadata** step described in section 2.2.

---

[15]SCORE2 risk score is obtained following the implementation available at: github.com/dvicencio/RiskScorescvd.

Table S9: Results for finetuning on lead I (upper table) and lead II (lower table) ECGs with confidence interval. AUROCs are reported for seven clinical tasks: LVEF refers to left ventricular ejection fraction; Dx denotes diagnosis; SubDx denotes subdiagnosis; SupDx denotes superdiagnosis; Form refers to waveform morphology; Rhyth denotes rhythm; Arrhy denotes arrhythmia.

| Dataset | MIMIC-IV | PTB-XL | | | | | Chapman |
|---|---|---|---|---|---|---|---|
| Task | LVEF | Dx | SubDx | SupDx | Form | Rhyth | Arrhy |
| **Lead I** | | | | | | | |
| *CLEF*-S (448K) | 0.8276±0.007 | 0.8473±0.011 | 0.8515±0.009 | 0.8311±0.007 | 0.7632±0.047 | 0.9515±0.011 | 0.9001±0.003 |
| *CLEF*-M (30.7M) | 0.8230±0.010 | 0.8499±0.011 | 0.8581±0.003 | 0.8335±0.005 | 0.7708±0.019 | 0.9415±0.006 | 0.9074±0.003 |
| *CLEF*-L (296M) | 0.7875±0.015 | 0.8533±0.005 | 0.8543±0.008 | 0.8320±0.004 | 0.7732±0.032 | 0.9442±0.007 | 0.9041±0.004 |
| **Lead II** | | | | | | | |
| *CLEF*-S (448K) | 0.8186±0.004 | 0.8284±0.008 | 0.8515±0.004 | 0.8475±0.003 | 0.7727±0.042 | 0.9510±0.011 | 0.9026±0.003 |
| *CLEF*-M (30.7M) | 0.8115±0.005 | 0.8361±0.004 | 0.8615±0.005 | 0.8482±0.004 | 0.7849±0.025 | 0.9531±0.011 | 0.9098±0.003 |
| *CLEF*-L (296M) | 0.7850±0.032 | 0.8333±0.009 | 0.8604±0.011 | 0.8489±0.007 | 0.7836±0.020 | 0.9538±0.005 | 0.9094±0.005 |

Table S10: Results for finetuning on 12-lead ECGs. AUROCs are reported, with numbers in brackets denoting the change relative to the corresponding single-lead baselines using lead I (in left hand side of Table 1), where (↑) and (↓) denote a performance gain and degradation, respectively. A grey background denotes that the method was outperformed by *CLEF*. LVEF refers to left ventricular ejection fraction; Dx denotes diagnosis; SubDx denotes subdiagnosis; SupDx denotes superdiagnosis; Form refers to waveform morphology; Rhyth denotes rhythm; Arrhy denotes arrhythmia.

| Model | MIMIC-IV | PTB-XL | | | | | Chapman |
|---|---|---|---|---|---|---|---|
| | LVEF | Dx | SubDx | SupDx | Form | Rhyth. | Arrhy. |
| ST-MEM | .8095 (↑.08) | .8340 (↑.23) | .8261 (↑.13) | .8459 (↑.11) | .6716 (↑.26) | .8419 (↑.12) | .8462 (↑.03) |
| KED | .8704 (↑.08) | .9293 (↑.15) | .9079 (↑.09) | .9250 (↑.10) | .8549 (↑.32) | .9683 (↑.11) | .9419 (↑.05) |
| Moment | .7884 (↓.02) | .8995 (↑.14) | .8727 (↑.12) | .8926 (↑.08) | .7874 (↑.28) | .9528 (↑.09) | .9298 (↑.04) |
| ECGFounder | .8748 (↑.05) | .9107 (↑.14) | .9106 (↑.10) | .9085 (↑.08) | .8046 (↑.18) | .9321 (↑.01) | .9362 (↑.07) |

## D SUPPLEMENTARY EVALUATIONS

### D.1 DETAILED CONFIDENCE INTERVAL RESULTS

This section supplements the results reported in section 4. To quantify uncertainty for the proposed *CLEF* model, we conducted 6 rounds of experiments with the prediction head initialized with different seeds '10, 42, 111, 123, 1111, 1234', and computed 95% confidence intervals for all reported metrics. Table S9 enumerates the results, where for the majority of the tasks, most confidence intervals remain tight $(0.003 - 0.011)$, apart from Form classification in the PTB-XL dataset, which shows larger variability across seeds. It can also be observed that the selected seed '42' yields relatively conservative results compared to other initializations.

### D.2 COMPARISON WITH FINETUNING ON 12-LEAD ECGS

To better quantify the performance upper bounds for downstream tasks, we compare single-lead results for *CLEF* against baseline methods that use full 12-lead ECG data. Table S10 summarizes the finetuning results for baseline methods on 12-lead ECGs. Several 12-lead baselines fail to surpass the performance of the single-lead *CLEF* model (shown as gray-highlighted cells). Performance changes relative to the corresponding single-lead baselines are shown in parentheses. All models benefited from multi-lead inputs for all tasks, except for Moment, where there is a small reduction in performance on the LVEF classification task when using 12-lead data. This confirms that additional leads provide valuable complementary information.

### D.3 LINEAR PROBING

This section supplements the linear probing experiments in section 4 in the main paper, where we analyzed the performance of *CLEF*-M linear probing against the baselines. Table S11 presents the detailed performance for *CLEF*-M and further results for *CLEF*-S and *CLEF*-L, comparing with the

Table S11: Results for linear probing on lead I ECG representations expressed as AUROCs. LVEF refers to left ventricular ejection fraction; Dx denotes diagnosis; SubDx denotes subdiagnosis; SupDx denotes superdiagnosis; Form refers to waveform morphology; Rhyth denotes rhythm; Arrhy denotes arrhythmia. Best result **bolded**. The second best underlined.

| Model | MIMIC-IV | PTB-XL | | | | | Chapman | Icentia11K | |
|---|---|---|---|---|---|---|---|---|---|
| | LVEF | Dx | SubDx | SupDx | Form | Rhyth | Arrhy | Beat | Rhyth |
| Moirai | .5000 | .4989 | .5000 | .5000 | .5000 | .5000 | .4988 | .5002 | .4927 |
| Moment | .7537 | .6392 | .6932 | .7611 | .5863 | .6227 | .7292 | .7298 | .6434 |
| ST-MEM | .5709 | .5652 | .5872 | .6189 | .5300 | .5374 | .6587 | .6640 | .5605 |
| KED | .7607 | .5935 | .6883 | .7514 | .5333 | .7372 | .7712 | .6984 | .7221 |
| *CLEF*-S | .7728 | .6005 | .6509 | .7503 | .5266 | .5872 | .6913 | .6581 | .6761 |
| *CLEF*-M | .7864 | .7256 | .7736 | .7903 | .6146 | .8077 | .7937 | .7302 | .7302 |
| *CLEF*-L | .7939 | .7388 | .7713 | .7971 | .6110 | .7966 | .8010 | .7843 | .7479 |
| ECGFounder | **.8228** | **.7945** | **.8423** | **.8443** | **.7412** | **.9422** | **.8428** | **.9692** | **.9721** |

Table S12: Results when retraining the backbone models with clinically-guided pretraining and finetuning, using lead I ECG. Numbers in brackets denote the change relative to the corresponding single-lead baselines using lead I, with (↑) denoting a performance gain and (↓) a degradation. LVEF refers to left ventricular ejection fraction; Dx denotes diagnosis; SubDx denotes subdiagnosis; SupDx denotes superdiagnosis; Form refers to waveform morphology; Rhyth denotes rhythm; Arrhy denotes arrhythmia. For better clarity, we highlighted cases where performance is better in teal.

| Model | MIMIC-IV LVEF | Dx | SubDx | PTB-XL SupDx | Form | Rhyth | Chapman Arrhy |
|---|---|---|---|---|---|---|---|
| ST-MEM | .7788 (↑ .00) | .7690 (↓ .01) | .7568 (↓ .01) | .7878 (↑ .01) | .5468 (↓ .04) | .8078 (↑ .07) | .8046 (↓ .01) |
| KED | .8267 (↓ .01) | .8729 (↑ .04) | .8569 (↑ .03) | .8330 (↑ .00) | .7481 (↑ .12) | .9067 (↑ .02) | .8956 (↑ .01) |
| ECGFounder | .8263 (↓ .03) | .8515 (↑ .01) | .8649 (↑ .02) | .8322 (↓ .01) | .7957 (↑ .04) | .9516 (↑ .00) | .9038 (↓ .01) |

baseline models. Among the 3 variants, *CLEF*-L achieves the highest linear probing performance with a 7.4% improvement over best baselines on average across 9 downstream classification tasks. The second best being *CLEF*-M (6.0% improvement), while *CLEF*-S exhibits a 7.3% performance deficit. This descending trend indicates that larger models develop more expressive and generalizable representations during pretraining, which can be effectively leveraged through simple linear classification heads.

Notably, comparing results in Table S11 with Table 1, *CLEF*-M generally achieves the best balance of capacity and data efficiency during fine-tuning, where it outperforms *CLEF*-L on many tasks, particularly those with limited labeled data. This potentially reflects a capacity–data mismatch under shared fine-tuning hyperparameters. The 296M-parameter *CLEF*-L requires stronger regularization and/or more labeled data for stable end-to-end fine-tuning. When it is given with relatively small downstream datasets (especially on wearable and ED tasks), it is more prone to overfitting or reaching optimization plateaus when using the same fine-tuning recipe as *CLEF*-M. Nevertheless, the stronger linear-probe performance of *CLEF*-L indicates that its representations are of higher quality.

Additionally, the supervised model ECGFounder outperformed all three variants of *CLEF*, with *CLEF*-M AUROCs an average of 12.7% lower than those of ECGFounder. This is expected, as ECGFounder was pretrained on a much larger labeled dataset, whereas *CLEF* relies purely on self-supervised learning.

### D.4 EFFECTIVENESS OF RISK-GUIDED PRETRAINING ON BASELINE MODELS

This section supplements the experiment in section 4 on applying our clinically-guided contrastive loss to the best semi-supervised baseline, KED. We further extend our analysis by applying the proposed pretraining approach to other baseline models, thereby testing its generalizability across architectures. Each baseline model has been initialized with its original pretrained weights and pretrained with our proposed clinically-guided pretraining approach. The AUROC results are presented in Table S12, with values in the brackets indicating the changes relative to the finetuning results of the original model weights, as reported in Table 1. On average, pretraining with our proposed clinically-guided method yields an overall performance gain of 0.7%. Among the three baselines, KED shows the largest improvement, with its AUROC score increasing by 3.0%. This substantial improvement likely stems from KED's reliance on purely self-supervised learning without explicit clinical guidance,

Table S13: Results on finetuning *CLEF* variants pretrained on the corresponding lead. The parentheses indicate the lead on which the model was pretrained. AUROCs are shown for Lead I (left) and Lead II (right). A teal text denotes improved performance compared to Table 1. LVEF refers to left ventricular ejection fraction; Dx denotes diagnosis; SubDx denotes subdiagnosis; SupDx denotes superdiagnosis; Form refers to waveform morphology; Rhyth denotes rhythm; Arrhy denotes arrhythmia.

| Task | MIMIC-IV LVEF | Dx | SubDx | PTB-XL SupDx | Form | Rhyth | Chapman Arrhy |
|---|---|---|---|---|---|---|---|
| *CLEF*$^{\text{I}}$-S | .8215 | .8536 | .8604 | .8324 | .7958 | .9449 | .9057 |
| *CLEF*$^{\text{I}}$-M | .8170 | .8505 | .8575 | .8336 | .7805 | .9476 | .9128 |
| *CLEF*$^{\text{I}}$-L | .7821 | .8606 | .8564 | .8327 | .7870 | .9513 | .9133 |

| Task | MIMIC-IV LVEF | Dx | SubDx | PTB-XL SupDx | Form | Rhyth | Chapman Arrhy |
|---|---|---|---|---|---|---|---|
| *CLEF*$^{\text{II}}$-S | .8207 | .8386 | .8620 | .8497 | .8001 | .9552 | .9107 |
| *CLEF*$^{\text{II}}$-M | .8102 | .8349 | .8719 | .8523 | .7921 | .9552 | .9135 |
| *CLEF*$^{\text{II}}$-L | .7862 | .8421 | .8680 | .8464 | .7829 | .9257 | .9134 |

Table S14: "Risk score" of CIFAR-100 superclasses and subclasses. Subclasses within the same superclass have similar risk scores, while more distant superclasses have larger dissimilar scores.

| Superclass | Subclass | Risk Score |
|---|---|---|
| Natural Environment (0.05–0.20) | Large natural outdoor scenes | 0.05 |
| | Trees | 0.10 |
| | Flowers | 0.15 |
| | Fruit and vegetables | 0.20 |
| Aquatic Life (0.25–0.30) | Fish | 0.25 |
| | Aquatic mammals | 0.30 |
| Invertebrates (0.35–0.40) | Insects | 0.35 |
| | Non-insect invertebrates | 0.40 |
| Land Mammals (0.45–0.70) | Small mammals | 0.45 |
| | Medium mammals | 0.50 |
| | Reptiles | 0.55 |
| | Large omnivores and herbivores | 0.60 |
| | Large carnivores | 0.65 |
| | People | 0.70 |
| Human-Made Objects (0.75–1.00) | Food containers | 0.75 |
| | Household furniture | 0.80 |
| | Household electrical devices | 0.85 |
| | Large man-made outdoor things | 0.90 |
| | Vehicles 1 | 0.95 |
| | Vehicles 2 | 1.00 |

making it particularly receptive to our clinically-guided pretraining strategy that provides essential domain-specific knowledge.

### D.5 SUPPLEMENTS IN PRETRAINING WITH A SPECIFIC LEAD

This section supplements section 4 on **pretraining on a specific lead**, which explores the upper bound performance by pretraining exclusively on lead I or lead II data, corresponding to the leads used in our downstream evaluation tasks. Although less generalizable, this method reduces potential domain shift compared with our primary pretraining approach using all 12 leads.

Table S13 present the results. The ones colored in red with an uparrow ($\uparrow$) denote a performance improvement compared to the performance of the corresponding-sized *CLEF* model pretrained on all 12 leads, while those with a downarrow ($\downarrow$) indicate a performance degradation. Overall, lead-specific pretraining yields performance gains across all tasks, with an average improvement of 2.0%. Specifically, *CLEF*$^{\mathrm{I}}$-S, *CLEF*$^{\mathrm{I}}$-M, and *CLEF*$^{\mathrm{I}}$-L achieve improvements of 3.4%, 1.6%, and 1.0%, respectively, compared to their corresponding models pretrained on all leads across all tasks and both leads. Interestingly, the **Small** variant gives the largest performance gain. This may be because the reduced capacity of the smaller model helps prevent overfitting on the relatively limited data, allowing it to capture the essential patterns more effectively than the larger counterparts.

### D.6 EFFECTIVENESS OF DIFFERENT LOSS COMPONENTS

An ablation study is conducted using CIFAR-100 (Krizhevsky et al., 2009) (the representative computer vision benchmark) to understand the contribution of different loss components. This dataset consists of 100 subclasses, each grouped into a higher-level superclass. While instances within the same superclass may belong to different subclasses, they are generally more similar to each other than to instances from different superclasses. To simulate the setup in informing the contrastive learning process with risk scores, we assign each CIFAR-100 superclass a risk score within the range $[0, 1]$. The assigned scores are decided in a way such that subclasses belonging to the same superclass receive relatively close risk scores, while subclasses from more distinct superclasses are assigned increasingly dissimilar scores. The detailed scores given to each subclass are provided in Table S14.

Table S15: Cosine similarity statistics between class pairs. A spade symbol (♠) indicates that the two classes belong to the same superclass in the CIFAR-100 dataset, whereas a diamond symbol (◇) represents pairs from different superclasses. Values are reported as mean ± standard deviation of cosine similarity computed over the sampled pairs. We also report the classification accuracy on both sub and superclass classification, which has 100 and 20 classes in total, with mean and standard deviation reported on 6 different random seed initializations. Best classification results are **bolded**.

| Class Pair | Groups | $\mathcal{L}_{nce}$ | $\mathcal{L}^w$ | $\mathcal{L}^d$ | $\mathcal{L}_{nce} + \mathcal{L}^d$ | $\mathcal{L}^w + \mathcal{L}^d$ |
|---|---|---|---|---|---|---|
| Palm – Pine Tree | ♠ | .058 ± .125 | .081 ± .139 | .699 ± .214 | .067 ± .132 | .556 ± .153 |
| Palm Tree – Baby | ◇ | .018 ± .060 | .025 ± .064 | .548 ± .131 | .016 ± .052 | .432 ± .075 |
| Pine Tree – Baby | ◇ | .012 ± .052 | .016 ± .053 | .564 ± .136 | .010 ± .045 | .438 ± .075 |
| Shark – Trout | ♠ | .022 ± .081 | .024 ± .078 | .738 ± .104 | .019 ± .077 | .542 ± .074 |
| Shark – Bicycle | ◇ | .014 ± .060 | .026 ± .067 | .527 ± .146 | .012 ± .051 | .385 ± .094 |
| Trout – Bicycle | ◇ | .023 ± .073 | .038 ± .081 | .501 ± .137 | .019 ± .062 | .380 ± .100 |
| Apple – Pear | ♠ | .053 ± .125 | .062 ± .148 | .761 ± .110 | .083 ± .162 | .559 ± .116 |
| Apple – Bottle | ◇ | .015 ± .055 | .017 ± .059 | .544 ± .107 | .030 ± .060 | .411 ± .081 |
| Pear – Bottle | ◇ | .036 ± .099 | .038 ± .094 | .589 ± .121 | .035 ± .083 | .452 ± .089 |
| Bridge – House | ♠ | .065 ± .129 | .087 ± .136 | .762 ± .135 | .057 ± .119 | .561 ± .115 |
| Bridge – Sea | ◇ | .044 ± .102 | .056 ± .113 | .311 ± .222 | .046 ± .107 | .284 ± .165 |
| House – Sea | ◇ | .018 ± .064 | .021 ± .067 | .307 ± .217 | .016 ± .062 | .262 ± .154 |
| Cloud – Forest | ♠ | .013 ± .052 | .013 ± .048 | .654 ± .215 | .011 ± .050 | .482 ± .133 |
| Cloud – Bus | ◇ | .015 ± .060 | .018 ± .055 | .220 ± .196 | .014 ± .053 | .201 ± .153 |
| Forest – Bus | ◇ | .033 ± .086 | .041 ± .091 | .357 ± .264 | .025 ± .074 | .275 ± .181 |
| *ResNet18* pretrained | | | | | | |
| Subclass (100) | | .621 ± .003 | .624 ± .005 | .635 ± .002 | .611 ± .004 | **.646** ± .006 |
| Superclass (20) | | .638 ± .004 | .639 ± .004 | .646 ± .003 | .627 ± .009 | **.658** ± .007 |

For the experiments, we pretrained the ResNet18 model (initialized with ImageNet-1K pretrained weights) using different loss functions, which are the standard contrastive loss $\mathcal{L}_{nce}$, the weighted contrastive loss $\mathcal{L}^w$, the dissimilarity alignment loss $\mathcal{L}^d$, $\mathcal{L}_{nce} + \mathcal{L}^d$, and our proposed $\mathcal{L}^w + \mathcal{L}^d$. The models were trained for 50 epochs with a batch size of 256, optimized using the AdamW optimizer with an initial learning rate of $1 \times 10^{-3}$. A Cosine Annealing learning rate schedule was used, and we stop the training when the training loss does not decrease over 10 epochs.

**Inter-class similarity analysis.** We first analyzed the similarity relationships between groups by randomly selecting 5 sets of triplet classes. Each triplet consisted of 2 subclasses from the same superclass and one subclass from a different superclass. For each triplet, we obtain the similarity scores between all pairs of samples (resulting in 10,000 scores per subclass pair, given 100 samples per subclass), and report the mean and standard deviation across groups. Results are summarized on the left of Figure S2 and detailed in the upper part of Table S15. It can be observed that training the model with only $\mathcal{L}_{nce}$ yields minimal distinction between similarity scores of samples within the same superclass (intra-class) compared to those from different superclasses (inter-class). When training with $\mathcal{L}^w$, the inter-class differentiation becomes marginally more pronounced, although it remains insufficiently distinct. Meanwhile, training with $\mathcal{L}^d$ produces substantially more distinguishable inter-class similarity differences. However, this loss function enforces hard mapping of inter-class distances to align with risk scores without constraining intra-class clustering, consequently increasing the intra-class similarity variance. This indicates that samples of the same class are scattered rather

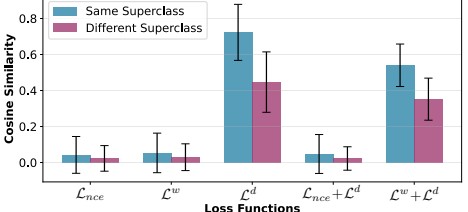 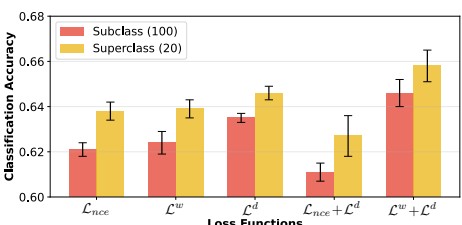

Figure S2: Ablation study on CIFAR-100 dataset. **Left:** Cosine similarity between samples from the same vs. different superclasses. Results averaged across 5 randomly selected triplet groups, with 2 of them from the same superclass. **Right:** Classification accuracy on CIFAR-100 subclass (100 classes) and superclass (20 classes) tasks.

Table S16: Effect of weighting over loss. All combinations are normalized by dividing by the sum of their coefficients to maintain comparable loss scales. Experiments were conducted over all classification tasks. For the 12-lead ECG datasets, we use lead I as input. Table cells are colored where darker blue indicates better performance and lighter blue indicates worse performance. Best results are **bolded**.

| Loss | MIMIC-IV | PTB-XL | | | | | Chapman | MUSIC | Icentia11K | | MC-MED | | |
| | LVEF | Dx | SubDx | SupDx | Form | Rhyth | Arrhy | Outcome | Beat | Rhyth | ED Dispo | DC Dispo | Acuity |
|---|---|---|---|---|---|---|---|---|---|---|---|---|---|
| $\mathcal{L}^d$ | 0.8051 | 0.7990 | 0.8135 | 0.8363 | 0.6451 | 0.9206 | 0.8808 | 0.4767 | 0.9803 | 0.9251 | 0.6016 | 0.6475 | 0.6222 |
| $\mathcal{L}^w + 5\mathcal{L}^d$ | 0.8096 | 0.8195 | 0.8536 | 0.8467 | 0.6917 | 0.9581 | 0.9028 | 0.4747 | 0.9780 | 0.9224 | 0.6070 | 0.6341 | 0.6179 |
| $\mathcal{L}^w + 2\mathcal{L}^d$ | 0.7965 | 0.8253 | 0.8518 | 0.8396 | 0.7365 | 0.9325 | 0.9088 | **0.5513** | 0.9755 | 0.9289 | 0.6148 | 0.6416 | 0.5861 |
| $\mathcal{L}^w + \mathcal{L}^d$ | 0.8083 | **0.8292** | **0.8566** | 0.8430 | 0.7409 | 0.9361 | 0.9061 | 0.5304 | 0.9792 | 0.9328 | **0.6397** | **0.6607** | **0.6510** |
| $\mathcal{L}^w + .5\mathcal{L}^d$ | **0.8204** | 0.8271 | 0.8503 | 0.8473 | **0.7471** | 0.9360 | **0.9130** | 0.5496 | **0.9858** | **0.9350** | 0.6026 | 0.6492 | 0.6221 |
| $\mathcal{L}^w + .2\mathcal{L}^d$ | 0.8118 | 0.8158 | 0.8520 | **0.8484** | 0.7325 | **0.9593** | 0.9032 | 0.4747 | 0.9794 | 0.9211 | 0.6082 | 0.6369 | 0.6216 |
| $\mathcal{L}^w$ | 0.8018 | 0.7990 | 0.8188 | 0.8387 | 0.6494 | 0.9215 | 0.8838 | 0.4701 | 0.9786 | 0.9241 | 0.6017 | 0.6446 | 0.6153 |

than forming compact clusters, which is a less desirable property for downstream classification. Meanwhile, the combination of $\mathcal{L}_{nce}$ with $\mathcal{L}^d$ offered limited gains compared to $\mathcal{L}_{nce}$. In contrast, our proposed combination $\mathcal{L}^w + \mathcal{L}^d$ achieves notable differentiation in similarity scores whilst maintaining consistency, which can be inferred from the reduced standard deviation.

**Classification accuracy on CIFAR-100.** We also provide classification accuracy over the 100 subclasses and also 20 superclasses, experiment initialized with 6 random seeds, which are '10, 42, 111, 123, 1111, 1234', and we present the average and standard deviation of the classification accuracy. Results are summarized on the right of Figure S2 and enumerated in the lower part of Table S15. For subclass classification (100 classes), our proposed $\mathcal{L}^w + \mathcal{L}^d$ combination achieves the highest accuracy of $64.6\% \pm 0.6\%$, outperforming all baseline approaches by at least $(1.7\%)$. Notably, whilst $\mathcal{L}^d$ alone yields the second best performance $(63.5\%\pm0.2\%)$, the combination of $\mathcal{L}_{nce} + \mathcal{L}^d$ exhibits degraded classification accuracy by $(3.8\%)$ $(61.1\% \pm 0.4\%)$, indicating that simply combining the InfoNCE loss with the difference alignment loss would contrarily harm the model performance, which is consistent with our similarity score observations. For superclass classification (20 classes), the performance remains consistent, with our proposed method achieving $65.8\% \pm 0.7\%$, bringing a substantial improvement by $(1.9\%)$ over the best-performance baseline $(64.6\% \pm 0.3\%)$ and by $3.0\%$ compared to the naïve contrastive baseline $(63.8\% \pm 0.4\%)$.

## D.7 ANALYSIS OVER DIFFERENT LOSS WEIGHTING

In Appendix D.6, we used CIFAR-100 to investigate how different combinations of losses influence the cosine similarity between representation pairs and classification accuracy. In this section, we extend this analysis to ECG tasks to examine the effect of having different weighting between the two proposed losses, $\mathcal{L}^w$ and $\mathcal{L}^d$. Specifically, we evaluated four configurations: (1) $\mathcal{L}^w$ only, (2) $\mathcal{L}^d$ only, (3) our default balanced weight $\mathcal{L}^w + \mathcal{L}^d$, and (4) unbalanced mixes with coefficients $\lambda = \{5, 2, 0.5, 0.2\}$ applied to $\mathcal{L}^w + \lambda\mathcal{L}^d$. All combinations have been normalized by dividing the sum of their corresponding coefficients to maintain comparable loss scales. It can be observed that *CLEF* is not particularly sensitive to the loss weight ratio (with an averaged standard deviation of 0.016 across all tasks), and the two components provide complementary guidance. Using balanced weights for $\mathcal{L}^w$ and $\mathcal{L}^d$ consistently provides strong performance, confirming our initial motivational analysis using CIFAR100.

## D.8 ANALYSIS OVER PRETRAINING PARAMETERS

**Effect of parameter $\alpha$:** We provide experiments on the effect of $\alpha$, using the medium variant of the *CLEF* model backbone, and pretrained the model with $\alpha = \{.1, .2, .3, .4, .5\}$. Results are enumerated in Table S17. The model remains robust when $\alpha$ is within the range $0.1 \check{} 0.3$, with 0.007 standard deviation averaged across all tasks. Increasing $\alpha$ beyond 0.3 results in a performance degradation, indicating that over-compressing the weighting factor can lead to suboptimal representation learning. Interestingly, if we compare the performance with Table 4, it can be observed that the performance when $\alpha = 0.5$ is similar to SimCLR, implying the weighting factor is over-compressed and becomes less effective. Overall, our results confirm that while $\alpha$ can affect the shape of the embedding space,

Table S17: Sensitivity analysis regarding $\alpha$. Experiments were conducted over all classification tasks. For the 12-lead ECG datasets, we use lead I as input. Table cells are colored where darker blue indicates better performance and lighter blue indicates worse performance. Best results are **bolded**.

| $\alpha$ | MIMIC-IV | PTB-XL | | | | | Chapman | MUSIC | Icentia11K | | MC-MED | | |
|---|---|---|---|---|---|---|---|---|---|---|---|---|---|
| | LVEF | Dx | SubDx | SupDx | Form | Rhyth | Arrhy | Outcome | Beat | Rhyth | ED Dispo | DC Dispo | Acuity |
| 0.5 | 0.8050 | 0.8084 | 0.8452 | 0.8357 | 0.7111 | 0.9212 | 0.8904 | 0.4737 | 0.9662 | 0.9381 | 0.6053 | 0.6112 | 0.5694 |
| 0.4 | 0.7983 | 0.8117 | 0.8550 | 0.8419 | 0.7291 | 0.9213 | 0.8950 | 0.4767 | 0.9550 | 0.9251 | 0.6019 | 0.6475 | 0.6407 |
| 0.3 | 0.8030 | **0.8295** | **0.8699** | **0.8515** | 0.7524 | 0.9338 | 0.9023 | **0.5513** | **0.9848** | **0.9475** | 0.6260 | 0.6491 | **0.6548** |
| **0.2** | 0.8083 | 0.8292 | 0.8566 | 0.8430 | 0.7409 | **0.9361** | 0.9061 | 0.5304 | 0.9792 | 0.9328 | **0.6397** | **0.6607** | 0.6510 |
| 0.1 | **0.8176** | 0.8203 | 0.8596 | 0.8484 | **0.7625** | 0.9360 | **0.9119** | 0.5476 | 0.9733 | 0.9397 | 0.6186 | 0.6593 | 0.6427 |

it does not directly drive the performance gains. The main improvements in the performance are due to using clinically-guided relative risk scores rather than the exact margin offset defined by $\alpha$.

**Effect of parameter** $\tau$ In our experiment, $\tau$ is set to 0.07 following prior work in contrastive learning (Wu et al., 2018; He et al., 2020). To further validate the sensitivity of the model to $\tau$ we conducted further experiments using the small and medium variants of the *CLEF* model backbone, with $\tau$ set to $\{0.01, 0.1, 0.5\}$. Results are enumerated in Table S18. Performance is best at small $\tau$ ($\tau = \{0.07, 0.1\}$) and degrades by little as $\tau$ increases. This indicates that lower $\tau$ enhances the discrimination between positives and negatives and is beneficial for learning clinical-informed ECG representations.

## D.9  COMPUTATIONAL COMPLEXITY ANALYSIS

The proposed clinically-guided pretraining framework introduces minimal computational overhead compared to the baseline SimCLR. Specifically, risk scores are computed once per patient from clinical metadata during the preprocessing stage, prior to model training. Following the SCORE2 algorithm detailed in Appendix C.6, each patient's risk score is derived from 7 clinical covariates through a series of operations (Equations S1–S3). For a single patient, this requires $O(1)$ operations, as the number of covariates and parameters is fixed.

Additionally, during training, the clinical guidance adds two operations per mini-batch: (i) obtaining weight matrix $\mathbf{W}_{i,k}$ from pairwise risk score dissimilarities $\mathbf{D}_{i,k}$ and missing metadata matrix $\mathbf{M}$, and (ii) computing the dissimilarity alignment loss $L^d$ (Eq. 7). Both operations require $O(B^2)$ computations for batch size $B$. In contrast, the standard SimCLR contrastive loss requires $O(B^2 d)$ operations for $d$-dimensional embeddings. Since for *CLEF*, $d \in \{256, 1024, 2048\}$, the additional $O(B^2)$ cost results in no significant increase in complexity.

## E  ALGORITHMS

Algorithm 1 summarizes our clinically-guided contrastive pretraining and downstream evaluation procedure. In the pretraining phase, we sample minibatches of ECG embeddings and generate stochastic augmentations, which are encoded by the foundation model $\mathcal{F}(\cdot)$. Risk scores from clinical metadata are used to compute pairwise dissimilarities, which are combined with multipliers weighted by the number of missing metadata to form the final pairwise weights. The foundation model $\mathcal{F}$ is updated by minimizing a combination of weighted contrastive loss $\mathcal{L}^w$ and dissimilarity alignment loss $\mathcal{L}^d$. In the evaluation phase, embeddings of downstream labeled ECGs are extracted and used to train a linear classifier or regressor $\mathcal{G}(\cdot)$. Model selection is performed by choosing the model

Table S18: Sensitivity analysis regarding $\tau$. Experiments were conducted over all classification tasks. For the 12-lead ECG datasets, we use lead I as input. Table cells are colored where darker blue indicates better performance and lighter blue indicates worse performance. Best results are **bolded**.

| $\tau$ | MIMIC-IV | PTB-XL | | | | | Chapman | MUSIC | Icentia11K | | MC-MED | | |
|---|---|---|---|---|---|---|---|---|---|---|---|---|---|
| | LVEF | Dx | SubDx | SupDx | Form | Rhyth | Arrhy | Outcome | Beat | Rhyth | ED Dispo | DC Dispo | Acuity |
| **.07** | **.8083** | .8292 | .8566 | .8430 | .7409 | .9361 | **.9061** | **.5304** | .9792 | **.9328** | **.6397** | **.6607** | **.6510** |
| .1 | .7809 | .8405 | **.8568** | **.8519** | **.7988** | **.9604** | .9024 | .4737 | .9440 | .9265 | .6033 | .6432 | .6463 |
| .5 | .7760 | **.8416** | .8440 | .8429 | .7742 | .9220 | .9022 | .4708 | .9345 | .9255 | .6038 | .6421 | .6308 |

that achieves the lowest validation loss. The final model is evaluated on the held-out test set using AUROC (classification) or MAE (regression).

## F    DISCUSSION OF CHOICE OF RISK SCORE

In this work, we used the SCORE2 risk score to guide contrastive learning. SCORE2 was selected because it is a standard cardiovascular risk score relevant to ECG data, which has been externally validated on data from over one million individuals from 15 countries, and its required input variables overlap with the metadata in the MIMIC-IV-ECG pretraining dataset. However, we do not claim that SCORE2 is the optimal risk score for *CLEF* as we did not test other risk scores. Furthermore, the optimal risk score might vary across applications. For instance, SCORE2 was developed using data from European subjects, for adults aged 40-69, and for long-term cardiovascular risk prediction. Alternative risk scores may be more appropriate for non-European populations, for different age groups, and for non-cardiovascular risk assessment. In addition, the optimal risk score will likely vary depending on the available metadata in a pretraining dataset.

## LLM USAGE STATEMENT

LLMs are used to aid in the writing of this paper. We use LLMs to improve the phrasing clarity to better convey our contributions and research findings to the readers. Specifically, this involves

---

**Algorithm 1:** Clinically-Guided Contrastive Pretraining and Evaluation

---

**Pretraining Phase:**

**Input:** Unlabeled ECG dataset $\mathbb{D} = \{(\mathbf{e}_s = (\mathbf{e}_s^1, \ldots, \mathbf{e}_s^{12}), \mathtt{a}_s)\}_{s=1}^N$, risk score function $\mathcal{R}(\cdot)$,

**Output:** Foundation ECG model $\mathcal{F}(\cdot)$

**for** *each minibatch* $\{\mathbf{e}_s^l\}_{s=1}^B$ **do**

    **for** *each* $\mathbf{e}_s^l$ **do**

        Sample stochastic augmentations $\mathbf{x}_i = \mathcal{T}(\mathbf{e}_s^l)$, $\mathbf{x}_j = \mathcal{T}(\mathbf{e}_s^l)$ ;

        Compute embeddings $\mathbf{z}_i = \mathcal{F}(\mathbf{x}_i)$, $\mathbf{z}_j = \mathcal{F}(\mathbf{x}_j)$ ;

        Obtain risk score $r_s = \mathcal{R}(\mathtt{a}_s)$ from clinical metadata (Eq. (3)) ;

    **for** *each pair* $(\mathbf{x}_i, \mathbf{x}_k)$ *in batch* **do**

        Compute risk dissimilarity $\delta_{ik} = (r^i - r^k)^2$ ;

        Normalize to $\mathbf{D}_{ik} \in [\alpha, 1]$ (Eq. (5)) ;

        Compute metadata missingness weight $\mathbf{M}_{ik}$ (Eq. (4)) ;

        Set final weight $\mathbf{W}_{ik} = \mathbf{D}_{ik} \cdot \mathbf{M}_{ik}$ ;

    Compute weighted contrastive loss $\mathcal{L}^w$ (Eq. (6)) ;

    Compute dissimilarity alignment loss $\mathcal{L}^d$ (Eq. (7)) ;

    Update encoder $\mathcal{F}(\cdot)$ by minimizing $\mathcal{L} = \mathcal{L}^w + \mathcal{L}^d$ ;

**Evaluation Phase:**

**Input:** downstream labeled dataset $\mathbb{C} = \{(\mathbf{z}_s = \mathcal{F}(\mathbf{e}_s^l), y_s)\}_{s=1}^M$

**Output:** downstream linear model $\mathcal{G}(\cdot)$

**for** *each labeled ECG* $(\mathbf{e}_s, y_s)$ *in* $\mathbb{C}_{train}$ **do**

    Extract embedding $\mathbf{z}_s = \mathcal{F}(\mathbf{e}_s)$ ;

    Train/update a linear classifier/regressor $\hat{y}_s = \mathcal{G}(\mathbf{z}_s)$ ;

**for** *each labeled ECG* $(\mathbf{e}_s, y_s)$ *in* $\mathbb{C}_{val}$ **do**

    Extract embedding $\mathbf{z}_s = \mathcal{F}(\mathbf{e}_s)$ ;

    Predict discrete/continuous values $\hat{y}_s = \mathcal{G}(\mathbf{z}_s)$ ;

Select $\mathcal{G}$ with best validation loss ;

**for** *each labeled ECG* $(\mathbf{e}_s, y_s)$ *in* $\mathbb{C}_{test}$ **do**

    Extract embedding $\mathbf{z}_s = \mathcal{F}(\mathbf{e}_s)$ ;

    Predict discrete/continuous values $\hat{y}_s = \mathcal{G}(\mathbf{z}_s)$ ;

Evaluate $\{\hat{y}_s\}$ against $\{y_s\}$ using AUROC (classification) or MAE (regression) ;

---

grammar correction, wording polishing, and sentence structure refinement. The extent of LLM assistance was minimal relative to the overall manuscript content. We note that LLMs are not involved in any aspect of data analysis, the development of methods, the design of the experiment, or results interpretation. The usage of LLMs is strictly limited to linguistic refinement. All scientific content originates entirely from the author's research work. The authors take full responsibility for all content presented in the paper and have verified that the usage of LLMs has not resulted in any form of plagiarism. All claims, findings, and contributions presented are the authors' own work and have been thoroughly validated for originality and accuracy.

