# OpenReview forum: "CLEF: Clinically-Guided Contrastive Learning for Electrocardiogram Foundation Models"
_ICLR.cc/2026/Conference — Submitted to ICLR 2026_

### Official Review · Reviewer_SYZy · 2025-10-29

**Soundness:** 2
**Presentation:** 3
**Contribution:** 2
**Rating:** 2
**Confidence:** 4

**Summary:**

The paper introduces CLEF (Clinically-Guided Contrastive Learning Framework), a self-supervised approach that integrates clinical risk scores into contrastive pretraining for ECG representation learning.
By weighting negative pairs and aligning embeddings with clinical dissimilarities, CLEF aims to encode clinically meaningful relationships within the latent space.
Extensive experiments across multiple ECG datasets demonstrate robust and consistent improvements over existing contrastive learning baselines.

**Strengths:**

1. Comprehensive and rigorous experimentation — The study presents extensive evaluations and ablation studies across diverse datasets and clinical tasks, clearly supporting the robustness of the proposed approach.

2. Clarity and strong organization — The methodology and motivation are well articulated with intuitive explanations and clear visualizations, making the paper easy to follow even for non-domain experts.

**Weaknesses:**

1. Lack of representation-level validation — Although the authors claim that CLEF adjusts contrastive representations based on clinical risk, the paper lacks direct empirical evidence demonstrating how this modification affects the underlying representation geometry. No analyses such as t-SNE or UMAP visualization, pairwise similarity distributions, or intra/inter-cluster distance comparisons are provided. All evaluations focus solely on downstream performance metrics, leaving the clinically guided nature of the learned representation as a theoretical assertion rather than an empirically verified outcome. Moreover, since CLEF introduces its clinical weighting only through additional loss terms atop a conventional contrastive backbone, potential representational shifts are likely to occur at the pairwise similarity level rather than in global embedding structure, making the claimed improvement difficult to confirm.

2. Limited clinical expressiveness due to single-lead input — The restriction to a single lead (Lead I) imposes inherent limitations on the model’s physiological depth. By excluding spatial context such as precordial lead variations and vectorcardiographic relationships, the model captures only waveform-level abstractions and neglects multi-lead correlations crucial for ischemic or regional cardiac representation. Consequently, the learned representation remains shallow in clinical richness despite its medical motivation.

**Questions:**

It would be valuable to visualize or quantify how the proposed clinical weighting actually reshapes the learned representation space, to provide more concrete evidence of the claimed effect.

---

> ### Author Response · Authors · 2025-11-23
> **Response to Reviewer SYZy**
>
> We thank the reviewer for acknowledging that our experiments are comprehensive and that the paper is well articulated.
>
> **Empirical evidence of the method:** While t-SNE and UMAP are widely used for 2D visualization of high-dimensional embeddings, they are not reliable tools for validating representation geometry. Both methods introduce distortions that undermine their use for empirical validation [1, 2]. Further, the outputs of t-SNE and UMAP are highly sensitive to hyperparameters and initialization, leading to inconsistent cluster shapes across runs and making any conclusions from a single visualization unstable. Moreover, both techniques can produce visually compelling but misleading clusters or patterns. t-SNE in particular tends to emphasize local groupings, sometimes “forcing” cluster structures even when none exist. Such projections can be manipulated to show arbitrary shapes, including those that resemble familiar objects, without preserving meaningful data properties. In summary, t-SNE and UMAP are useful for qualitative exploration, but they are not valid tools for empirically verifying the structure of learned representations.
>
> Our claims about the nature of CLEF's learned representations are based on extensive empirical evaluations across a wide range of unseen downstream tasks. CLEF was pretrained without access to any of the evaluation datasets and tested on 18 clinical tasks across 7 held-out datasets spanning different pathologies, devices, and acquisition settings. This rigorous out-of-distribution benchmarking, including classification and regression tasks from sources such as PTB-XL, MIMIC-IV, Chapman, and Aurora BP, consistently demonstrates that CLEF outperforms strong self-supervised and supervised baselines. The robustness of these results (across leads, tasks, and model sizes) provides concrete evidence that CLEF captures clinically meaningful structure.
>
> Nevertheless, we have now visualised the latent space produced by CLEF using t-SNE visualisation ([anonymised link to the figure](https://anonymous.4open.science/r/image-1626/tsne_risk_score.png)). The visualization reveals a coarse, continuous distribution of risk scores across the embedding space, ranging from higher risk scores in the upper-left corner to lower risk scores in the lower-right corner. This demonstrates that CLEF effectively guides the model to produce embedding spaces that reflect the input risk score. There are no distinct clusters, which we believe is as expected because: (i) risk scores are numerical, implying a continuous distribution rather than discrete categories; (ii) Multiple pathologies can lead to similar risk scores but have varying effects on the ECG (e.g., acute hypertension vs. chronic hypertension alongside left ventricular hypertrophy); and (iii) Missing metadata introduces imprecision in the risk scores, contributing to less distinct boundaries in the embedding space.
>
> - [1]: Stop Misusing t-SNE and UMAP for Visual Analytics, https://arxiv.org/abs/2506.08725
> - [2]: How to Use t-SNE Effectively, https://distill.pub/2016/misread-tsne
>
> **Clinical value of single lead ECG**: We agree with the reviewer that a single-lead ECG captures less information than a full 12-lead recording. We respectfully clarify that this work does not claim single-lead ECG achieves the clinical expressiveness of 12-lead recordings. In fact, we acknowledged the loss of spatial information (lines 43-44) and empirically show in Tbl. S10 that all baseline models benefited from multi-lead inputs for all tasks, except for Moment. The reason we choose to exclusively focus on single-lead ECG is that there has been an increased usage of clinical-grade and consumer wearables. Single-lead ECG has also been used for practical diagnoses, especially in wearable contexts. Moreover, compared to in-clinic 12-lead ECG, single-lead ECG also enables continuous monitoring beyond clinical settings, supporting early cardiac event detection, long-term tracking, and proactive interventions (lines 11-13, lines 40-47).
>
> Our core idea of this study is that when embedding geometry reflects similarities under a chosen risk score $\mathcal{R}(\cdot)$, it improves the performance of a generalisable foundation model over downstream diagnostic tasks, which potentially compensates for the information loss compared to a 12-lead ECG. Empirical results also show that our single-lead CLEF model outperforms several 12-lead baselines (Tbl. S10 and Appendix D.2).

---

### Official Review · Reviewer_ziBe · 2025-10-30

**Soundness:** 3
**Presentation:** 4
**Contribution:** 3
**Rating:** 6
**Confidence:** 4

**Summary:**

**Summary**

This paper proposes a novel framework called CLEF (Clinically-Guided Contrastive Learning) for electrocardiogram (ECG) foundation models. Its core innovation lies in overcoming the limitations of conventional self-supervised pretraining methods that fail to incorporate clinical domain knowledge and treat all negative pairs equally. CLEF dynamically links the weighting of negative pairs in contrastive learning to the clinical dissimilarity between subjects—measured via an established clinical risk score. This allows the model to align the ECG embedding space with clinically meaningful differences, performing stronger repulsion for pairs with high risk dissimilarity and weaker repulsion for clinically similar pairs. The framework incorporates a *negative weighting loss ($\mathcal{L}^w$)* and a *dissimilarity alignment loss ($\mathcal{L}^d$)*, along with a mechanism for missing metadata, to optimize the model for 18 downstream clinical classification and regression tasks.

**Strengths:**

- **Problem orientation is concrete and clinically motivated.** The work targets label scarcity and weak cross-device/population generalization in single-lead ECG, injecting domain knowledge via routinely captured clinical metadata rather than dense manual labels.

- **Method is technically neat and conceptually coherent.** Clinical dissimilarity is converted into a training signal through risk-aware negative reweighting, paired with a distance-alignment term, while missing metadata are modeled explicitly instead of being discarded.

- **Pretraining is scalable and evaluation is broad.** Models are trained on large ECG corpora and assessed across diverse datasets and tasks (classification and regression), including cross-dataset transfer, strengthening external validity.

- **Performance gains are consistent against strong baselines.** The approach surpasses competitive self-supervised and supervised methods on standard metrics, with improvements that persist across tasks, splits, and evaluation settings.

- **Robustness is carefully documented.** Multiple random seeds with confidence intervals and component-wise ablations delineate the contribution of each design choice and indicate stable training behavior.

**Weaknesses:**

- **Teacher–task mismatch (SCORE2 for immediate diagnostics).** SCORE2 supervises with a 10-year cardiovascular risk (prognosis) derived from demographics and vitals, yet the paper applies it to pretrain models for short-horizon diagnostic tasks (e.g., arrhythmia/rhythm/beat classification) without justifying why a long-term risk surrogate is an appropriate teacher for immediate signal decisions. A minimally necessary control is to compare against an *acute* risk or diagnosis-proximal target.



- **Unproven hyperparameter choices on ECG data.** The paper defines the minimum-distance parameter \alpha and the temperature \tau, and fixes the final objective to L=L_w+L_d, but provides no ECG-domain sensitivity for \alpha, \tau, or the 1:1 weighting—pointing instead to an image-domain proxy (CIFAR-100) to justify loss composition. This leaves robustness and optimality on ECG tasks unsubstantiated.



- **Counter-intuitive missing-metadata multiplier** M_{ik}**.** By construction, M_{ik}=\exp\!\big(-\frac{A-m_i}{A}\times\frac{A-m_k}{A}\big) downweights pairs with more complete metadata and increases weights as metadata go missing—an opaque choice with no principled motivation given, despite being a core mechanism.



- **Missing-metadata handling shows mixed efficacy.** The ablation on Table 5 reveals that removing the missing-metadata mechanism  sometimes improve results, indicating the component is not uniformly beneficial and may be suboptimal for certain tasks/datasets. This contradicts a claim of universal benefit and warrants per-task analysis.



- **Weighted-InfoNCE can suffer batch-composition dominance.** Because negatives are reweighted inside the denominator of the contrastive loss, large-\Delta r pairs can dominate gradients, risking reduced representation uniformity and batch-effect brittleness—especially with small batches or skewed risk distributions. The paper does not probe these interactions.

**Questions:**

1. What is the empirical correlation between SCORE2 differences and the target downstream tasks (e.g., arrhythmia classification, blood pressure prediction)? This would validate the core premise.

2. For missing metadata handling, why not use imputation or domain adaptation instead of multiplicative weighting? Have alternatives been explored?

3. Why does CLEF-L underperform CLEF-M consistently? Is this due to optimization dynamics specific to large models on ECG?

4. In Table S13, lead-specific pretraining outperforms random-lead pretraining (2% avg gain). How do you reconcile this with the claimed advantage of random lead selection?

5. For ECGFounder comparison: it uses 150 diagnostic categories during supervision. How much of the performance gap is attributable to labeled data vs. method differences?

---

> ### Author Response · Authors · 2025-11-23
> **Response to Weaknesses (Part I)**
>
> We thank the reviewer for the constructive feedback, and for highlighting the novelty, clinical motivation, strong performance, and evaluation robustness of CLEF. We address the reviewer’s questions with explicit pointers to sections and tables in the paper.
>
> **The usage of SCORE2**: We emphasize that CLEF is designed to incorporate a clinical risk score $\mathcal{R}(\cdot)$ that maps subject-level metadata to the likelihood of a health outcome. SCORE2 is one instantiation of this general concept, which is a cardiovascular risk score validated on more than 600,000 individuals and more than 30,000 cardiovascular events. Our core idea is that when embedding geometry reflects similarities under a chosen risk score $\mathcal{R}(\cdot)$ (which is not a specific diagnosis label), it improves the performance of a generalisable foundation model over downstream diagnostic tasks.
>
> We chose to use information relating to the underlying cardiovascular state for model pre-training (in our case, a long-term risk score), rather than information relating to particular short-horizon diagnoses. We anticipated that this would enable the development of generalisable models since long-term risk factors (e.g. hypertension, diabetes, hyperlipidemia, and age) are also risk factors for short-horizon diagnoses (e.g. structural heart disease, left ventricular dysfunction, ischemia, and arrhythmias). Therefore, SCORE2 is not used as a surrogate for arrhythmia label; it is used to organize the embedding space along an axis of overall cardiovascular disease burden, which is mechanistically linked to both structural and rhythm abnormalities that our downstream tasks probe.
>
> Additionally, it would have key advantages over using information relating to short-horizon diagnoses: (i) CLEF can be used with datasets containing only patient-level metadata rather than requiring manual annotations of individual ECGs; and (ii) potentially, CLEF could result in more generalisable models than those trained with particular short-horizon diagnoses. We compared the performance of CLEF models with those based on the ECGFounder foundation model, which was trained with short-horizon diagnoses. We found that in some circumstances, CLEF models performed comparably to ECGFounder (e.g. when pretrained only on lead-I data for classification tasks), demonstrating the potential value of CLEF.
>
> We appreciate the reviewer’s suggestion to compare against an acute or diagnosis-proximal risk score. However, the metadata available in MIMIC-IV-ECG are limited (Tbl. S2), and this constrains the construction of any validated acute-risk surrogate. In particular, variables commonly required for short-horizon risk scores, such as cardiac enzymes, hemodynamic instability markers, or time-varying physiological trends, are not available at the subject level in this dataset. MIMIC-IV-ECG is, nevertheless, the largest public ECG corpus with routinely captured metadata sufficient to compute at least one clinically validated risk score (SCORE2), which makes it uniquely suited for evaluating CLEF in an open, reproducible setting. While practitioners with access to private datasets containing richer metadata could apply CLEF with domain-appropriate acute-risk functions, this is not feasible in our context. To acknowledge this broader space, we have added Appendix F outlining alternative risk scores and discussing directions for future work. We view the systematic identification of the optimal risk function for contrastive pretraining as an important but orthogonal research problem that merits a dedicated investigation.

---

> ### Author Response · Authors · 2025-11-23
> **Response to Weaknesses (Part II)**
>
> **The effect of parameter $\alpha$**: We appreciate the reviewer’s observation. Our core experiments (Tbls. 1 and 2, and S9) has indicate the stable performance of CLEF and its robustness to the choice of $\alpha$ (which defines the minimum separation enforced between positives and negatives). To make this explicit, we have now conducted an ablation (now added to Appendix D.8 and Tbl. S17), where $\alpha$ is varied across \{.1, .2, .3, .4, .5\}. We observed: (1) The model remains robust when $\alpha$ is within the range .1–.3, with .007 standard deviation averaged across all tasks. (2) Increasing $\alpha$ beyond .3 results in a slight performance drop, indicating that over-compressing the weighting factor can lead to model collapse. (3) Interestingly, if we compare the performance with Tbl. 4, it can be observed that the performance when $\alpha=.5$ is similar to SimCLR, implying the weighting factor is over-compressed and becomes less effective. Overall, our results confirm that while $\alpha$ can affect the shape of the embedding space, it does not directly drive the performance gains. The main improvements in the performance are due to using clinically-guided relative risk scores rather than the exact margin offset defined by $\alpha$. The updated ablation study indicates that the value of $\alpha=.2$ which was used in the remaining studies is appropriate.
>
> **Effect of parameter $\tau$**: In our experiment, $\tau$ is set to $0.07$ following prior work in contrastive learning [1,2]. To further validate the sensitivity of the model to $\tau$ we conducted further experiments using the small and medium variants of the CLEF model backbone, with $\tau$ set to \{0.01, 0.1, 0.5\}. Results are added to Tbl. S18 in the revised version. Performance is best at small $\tau$ ($\tau=$ {0.07, 0.1}) and degrades by little as $\tau$ increases. This indicates that lower $\tau$ enhances the discrimination between positives and negatives[3] and is beneficial for learning clinical-informed ECG representations.
>
> [1] Unsupervised Feature Learning via Non-Parametric Instance Discrimination, CVPR 2018
> [2] Momentum Contrast for Unsupervised Visual Representation Learning, CVPR 2020
> [3] Understanding the Behaviour of Contrastive Loss, CVPR 2021
>
>
> **Effect of contribution of $\mathcal{L}^w$ versus $\mathcal{L}^d$**:
> To clarify on the functional roles of our two losses, we initially performed preliminary experiments on images, as a domain-agnostic starting point (results not shown with CIFAR100 using different weighting). To further confirm our choice of having equal weight over $\mathcal{L}^w$ and $\mathcal{L}^d$, we evaluated four configurations: (1) $\mathcal{L}^w$ only, (2) $\mathcal{L}^d$ only, (3) our default balanced weight $\mathcal{L}^w + \mathcal{L}^d$, and (4) unbalanced mixes with coefficients $\lambda = \{5, 2, .5, .2\}$ applied to $\mathcal{L}^w + \lambda\mathcal{L}^d$. The results are now added to Tbl. S16 in the revised version. These results show that CLEF is not particularly sensitive to the loss weight ratio (with an averaged standard deviation of .016 across all tasks), and the two components provide complementary guidance. Using balanced weights for $\mathcal{L}^w$ and $\mathcal{L}^d$ consistently provides strong performance, confirming our initial motivational analysis.

---

> ### Author Response · Authors · 2025-11-23
> **Response to Weaknesses (Part III)**
>
> **Handling missing-metadata with $\mathbf{M}$**: Our weighting mechanism is designed to account for the reliability of the metadata used to compute SCORE2. When metadata is complete and the risk score is therefore reliable, the corresponding entries in $\mathbf{M}$ reduce the dynamic range of the negative weights. This allows the dissimilarity term to more directly govern which negatives should be emphasized or de-emphasized. In contrast, when many metadata variables are missing, and SCORE2 must rely heavily on default assumptions, the resulting risk differences are less trustworthy. In these cases, the corresponding $\mathbf{M}$ values move the weighting closer to uniform SimCLR-like behavior, preventing uncertain or noisy risk differences from exerting undue influence on the contrastive objective. Importantly, higher values in $\mathbf{M}$ do not reflect greater trust in the risk score; instead, they indicate that we are less willing to amplify or suppress a negative pair based on an unreliable SCORE2 estimate. Because all samples share some degree of metadata missingness within the same corpus, $\mathbf{M}$ functions as a relative reliability adjustment (and not a penalty on complete metadata) and ensures that only well-supported risk differences meaningfully shape the embedding geometry. We have clarified this interpretation in the revised manuscript in Sections 2.2 and 2.3 to avoid the impression that the method up-weights low-quality metadata or rewards missingness.
>
> **Efficacy of handling missing metadata**: We acknowledge that the proposed method in handling missing data does not always improve performance; however, it does most of the time. We respectively note that we did not claim the weight matrix that handles missing metadata to have a universal benefit. Rather, it serves as a relative adjustment to account for the less reliable risk score due to the absence of metadata. We've further clarified this in Section 4. In general, adding the module of handling missing metadata improved performance for 24/30 tasks across all 3 variants of CLEF, with AUROC scores decreased by $1.9\%$ when removed. Additionally, it shows a larger decrease with wearable datasets (4.0\% drop), suggesting that these tasks are more sensitive to missing-metadata issues and thus benefit more from explicitly modeling such cases.
>
> **Effect of small batch**: The pairwise dissimilarities $D_{ik}$ are normalized to lie in $[\alpha, 1]$ using the batch-wise min and max of $\delta_{ik} = (r_i - r_k)^2$. With our $\alpha = 0.2$, even risk-similar negatives have $D_{ik} \ge 0.2$. Multiplying by $M_{ik} \in (0.83,1]$ (this is due to we only have 3/7 available metadata from MIMIC-IV dataset for SCORE2 calculation) yields weights $W_{ik} \in [\alpha e^{-1},\, 1] \approx (0.164,\, 1]$. Thus, the biggest weight can be at most $\sim 6\times$ larger than the smallest ones; there is no possibility of unbounded gradient dominance. Moreover, the dissimilarity alignment loss $\mathcal{L}^d$ does not involve batch-normalized weights. This smoothes and spread-out, embeddings, counteracting any tendency for a few negatives to dominate the InfoNCE term.
>
> In addition, our robustness to such an effect is empirically validated in three ways: (1) Tbl. S9 reports confidence interval results across 5 seeds for CLEF variants on 7 classification tasks. Standard deviations are small (e.g., CLEF-S lead-I LVEF: $0.8276 \pm 0.007$; PTB-XL Dx: $0.8473 \pm 0.011$; Chapman arrhythmia: $0.9001 \pm 0.003$), indicating stable training without batch-dependent. (2) CLEF-S/M/L all show consistent improvements over baselines across 18 tasks and 7 datasets, including cross-dataset transfer and wearable signals. If batch composition or skewed risk distributions were causing brittle behavior, we would expect much larger variance and inconsistent gains, which we do not observe. (3) Applying the clinically guided objective on top of three different pre-trained FMs (KED, ST-MEM, Moirai, Moment) yields small but consistent AUROC improvements (0.7\% on average; 3.0\% for KED), again without training instabilities.
>
> In general, we note that perfect uniformity is not the goal in this medical setting. We intentionally deviate from the "maximally uniform" regime of standard InfoNCE to structure the embedding space according to clinically meaningful differences. The bounded weights and the auxiliary alignment loss $L^d$ ensure that this structuring does not collapse the representation or make it overly dependent on a few extreme-risk examples. Additionally, in our experiments, we use a batch size of 64, which has been a common choice in contrastive learning [1,2,3]. Our empirical results demonstrate the effectiveness.
>
> - [1] SimCSE: Simple Contrastive Learning of Sentence Embeddings
> - [2] Contrastive Cross-Modal Learning for Infusing Chest X-ray Knowledge into ECGs
> - [3] PaPaGei: Open Foundation Models for Optical Physiological Signals

---

> ### Author Response · Authors · 2025-11-23
> **Response to Questions (Part I)**
>
> **Correlation between SCORE2 and downstream**: We were unable to compute a correlation coefficient between SCORE2 and each downstream label because our downstream datasets do not contain enough metadata required to calculate SCORE2. Nevertheless, our results offer strong evidence that SCORE2 differences capture structure relevant across a wide range of downstream tasks. First, across 13 downstream classification tasks (including LVEF, PTB-XL Dx/SubDx/SupDx/Form/Rhythm, Chapman arrhythmia, MUSIC SCD, Icentia11K beat and rhythm, and ED/DC disposition and acuity), CLEF-M improves average AUROC by 3.1\% over the strongest baseline FM when pretrained on 12-lead ECG and evaluated on single-lead inputs. Second, for morphology- and rhythm-driven tasks (PTB-XL Form/Rhythm and Chapman arrhythmia), where ECG structure is heavily shaped by underlying cardiovascular pathology, CLEF’s best variant improves AUROC by 5.9\% on lead I and 8.5\% on lead II relative to the strongest semi-supervised FM. Third, for longer-horizon or risk-proximal tasks such as sudden cardiac death (MUSIC), ED/DC disposition, and triage acuity, CLEF achieves a 2.5\% mean AUROC improvement over the best baseline. Fourth, on regression tasks (MIMIC-IV LVEF and two BP datasets), CLEF reduces MAE by 2.9–3.2\% compared with the strongest FM baseline. These gains are largest on tasks that are conceptually closest to global CVD risk (SCD, LVEF, BP, ED/triage outcomes), but remain substantial on purely morphological and arrhythmia tasks. This pattern indicates that SCORE2-guided geometry aligns with broad, clinically meaningful ECG variation rather than overfitting to a narrow endpoint. Notably, if SCORE2 were strongly correlated with all tasks, we would expect overfitting to risk factors rather than improved generalisation. Instead, the consistent improvements across heterogeneous tasks demonstrate that SCORE2 provides a valuable, domain-informed prior that enhances the learning of generalisable ECG representations.
>
> **Alternatives to handling missing data**: We imputed missing values (lines 1420–1427) only to provide the model with a minimal baseline for risk score computation. However, these values are inherently guesses and therefore less reliable. Thus, $\mathbf{M}$ is introduced to adjust the weighting during pretraining. Domain adaptation expects structured and consistent domain shifts, while metadata is patient-specific. Thus, we consider domain adaptation not suitable.
>
> **Performance difference between CLEF-M and CLEF-L**:
> CLEF-S/M/L differ in their learning capacity (448k, 30.7M, 296M parameters). All three are pretrained on the same 161,352-patient MIMIC-IV-ECG cohort with identical hyperparameters (learning rate, batch size, epochs), and fine-tuned on the same downstream splits. As reported in Appendix D.3, CLEF-L achieves the strongest linear-probe performance: a 7.4\% average AUROC improvement over the best baseline across 9 classification tasks, compared with 6.0\% for CLEF-M and a 7.3\% deficit for CLEF-S. This suggests that the large model indeed learns more expressive and generalizable representations during pretraining. On fine-tuning, CLEF-M generally achieves the best balance of capacity and data efficiency: it outperforms CLEF-L on many tasks, particularly those with limited labeled data. Our interpretation is that this is not a failure of the pretraining objective, but a consequence of capacity–data mismatch under shared fine-tuning hyperparameters. The 296M-parameter CLEF-L requires stronger regularization and/or more labeled data for stable end-to-end fine-tuning; given the relatively small downstream datasets (especially on wearable and ED tasks), it is more prone to overfitting or optimization plateaus when using the same fine-tuning recipe as CLEF-M. The stronger linear-probe performance of CLEF-L indicates that its representations are higher quality; the gap primarily reflects downstream optimization rather than pretraining instability. We have further clarified this in Appendix D.3.

---

> ### Author Response · Authors · 2025-11-23
> **Response to Questions (Part II)**
>
> **Random lead selection in pretraining**: Our design choice of using random lead selection during pretraining provides a single foundation model that can be fine-tuned to any lead (I, II, or others), which is important because different wearable devices use different leads (e.g., chest electrodes vs. smartwatch lead-I approximations), and the position of the target lead may not be known at pretraining time. In Appendix D.5, we explore lead-specific pretraining as an upper bound: we pretrain exclusively on lead I or II for 10 epochs (shorter due to reduced data diversity) and then fine-tune on the corresponding lead. Tbl. S13 and Fig. 4 trade-offs show that (1) Random-lead pretraining: single model, directly applicable to any lead, small performance cost ($\sim$ 2\% AUROC) relative to an oracle that knows the target lead during pretraining. (2) Lead-specific pretraining: slightly higher per-lead performance, but requires a separate FM per lead and is less flexible for cross-device deployment. Our claims about the advantage of random lead selection refer to this flexibility and generalization across leads, not to a claim that it strictly dominates lead-specific pretraining in per-lead AUROC. We have made this distinction explicit in Section 3
>
> **Performance gap between the supervised baseline**: We compare CLEF to ECGFounder using the same backbone family (ResNeXt1D variants) and the same downstream evaluation pipeline: ECGFounder is trained in a fully supervised manner on more than 10M labeled ECGs with 150 diagnostic categories. CLEF is trained self-supervised on 161,352 MIMIC-IV-ECG subjects using only metadata-derived SCORE2 risk scores (with missingness and imputation as described above), without any ECG-level labels. Despite this large gap in label volume and pretraining dataset size, we find: (1) CLEF-M’s AUROCs are on average 12.7\% lower than those of ECGFounder under fine-tuning, which we explicitly note in Appendix D.3. (2) When we restrict pretraining to lead I only (thereby better matching ECGFounder’s lead setting), CLEF becomes comparable to ECGFounder on several tasks, closing much of the gap without using ECG labels. Given that the backbone family is similar, CLEF improves other FMs when applied as an additional pretraining stage (Tbl. S12), and CLEF uses no ECG labels while ECGFounder uses large-scale supervision, we believe the dominant contributor to the performance gap is the massive labeled dataset rather than an inherent disadvantage of our proposed objective.

---

### Official Review · Reviewer_YTHd · 2025-11-01

**Soundness:** 4
**Presentation:** 4
**Contribution:** 4
**Rating:** 6
**Confidence:** 2

**Summary:**

This paper presents CLEF (Clinically-Guided Contrastive Learning), a self-supervised framework for ECG foundation models that integrates clinical risk scores into the contrastive pretraining process. The key idea is to move beyond traditional binary notions of similarity/dissimilarity and use continuous, clinically meaningful dissimilarities derived from metadata (e.g., SCORE2 cardiovascular risk) to guide the representation learning. The method introduces two loss components: a weighted contrastive loss (Lw), which adaptively weights negative pairs based on clinical risk and missing metadata, and a dissimilarity alignment loss (Ld), which aligns embedding similarity with clinical risk distance. CLEF is pretrained on 161K 12-lead ECGs from MIMIC-IV, and evaluated across 18 tasks spanning 7 datasets. The results show strong improvements over both self-supervised and foundation model baselines (up to +3.1% AUROC), and competitive performance with the supervised ECGFounder model.

**Strengths:**

The idea of weighting negative pairs using risk score–derived dissimilarities is elegant and well-motivated. It directly connects the latent geometry of embeddings to real-world clinical semantics, improving both interpretability and utility.
Strong empirical evaluation and broad benchmarking.
CLEF achieves strong performance even when pretrained on 12-lead data but fine-tuned on single-lead (lead I/II) tasks, suggesting it can effectively bridge clinical and consumer-grade ECG domains — a valuable property for real-world applications.
The framework (Figure 1) is well-illustrated, and the mathematical formulation of each component (Eqs. 5–7) is consistent and interpretable. The model’s connection to clinical reasoning adds conceptual clarity often missing in foundation model research.

**Weaknesses:**

While SCORE2 is a standard cardiovascular risk score, it is not necessarily the optimal or most generalizable measure for all datasets or populations (especially non-European cohorts).
Since metadata such as age, gender, and blood pressure are correlated with many downstream labels, the inclusion of such information during pretraining might inadvertently leak label information.
Although the weighting matrix W = D⊙M is central to the approach, the paper does not deeply explore sensitivity to α (Eq. 5) or the relative contribution of Lw vs Ld. This makes it difficult to assess how robust the improvement is to hyperparameter tuning.
The authors claim CLEF embeddings reflect “clinically meaningful geometry,” but there is no direct visualization or correlation analysis between latent distances and clinical outcomes. A t-SNE or UMAP projection aligned with risk scores would be valuable.
Computational cost and scalability discussion is missing.

**Questions:**

Could this clinically-guided weighting be applied to other biosignals (EEG, PPG, or medical imaging) with analogous metadata?
Did you test other risk scoring systems beyond SCORE2? For example, Framingham or ASCVD risk models — to verify if the method’s gains are score-agnostic.
What is the clinical interpretability of embeddings?
Have you examined whether clusters in the latent space correspond to specific cardiac pathologies or risk strata?
Does the model’s advantage persist under domain shift?
For example, does pretraining on MIMIC-IV generalize equally well to wearable ECG datasets with different sampling rates and demographic distributions?

---

> ### Author Response · Authors · 2025-11-23
> **Response to Weaknesses**
>
> We thank the reviewer for recognizing that our method is well-motivated, supported by extensive experiments, and clearly illustrated. Below, we provide detailed responses to the reviewer's questions, including references to the relevant revised sections of the paper.
>
> **The use of SCORE2**: We thank the reviewer for highlighting that SCORE2 might not be a globally optimal measure. In this work, we selected the SCORE2 risk score as a principled and clinically interpretable source of similarity information during pretraining because (1) it is clinically validated at scale, and (2) it incorporates commonly available metadata variables available in MIMIC-IV-ECG. We have now clarified in Section 2.2 that it was developed and validated using data solely from European populations and that this might limit its generalisability. We also added Appendix F, in which we discuss the choice of the SCORE2, its limitations, and considerations when selecting risk scores for future applications of CLEF.
>
> **Potential Label Leakage**: We agree with the reviewer that demographic and clinical metadata (age, gender, systolic blood pressure) are often correlated with downstream clinical labels. However, this reflects real-world physiology that the model is intended to capture, and the inclusion of such information is necessary to produce clinically meaningful representations. Importantly, we evaluate downstream tasks e.g., LVEF and diagnostic classification, which do not have a direct connection to these metadata. CLEF mitigates leakage in 3 ways: (1) We do not train CLEF to predict or reconstruct SCORE2, nor do we encode SCORE2 directly. Instead, SCORE2 induces a soft ordering over negative pairs (Eq. 6, Section 2.3). This shapes the embedding space without embedding metadata values themselves, and reduces the risk of label leakage compared to approaches that predict age, gender, or full metadata vectors. (2) Metadata is used only during pretraining and never provided as input at inference. Any advantage must come from improved ECG representations, rather than direct access to the covariates. (3) Additionally, our mechanism for handling missing metadata prevents the model from collapsing into metadata-only similarities and encourages the model to use ECG morphology.
>
> **Effect of parameter $\alpha$**: We appreciate the reviewer’s observation. Our core experiments (Tbls. 1 and 2, and S9) has indicate the stable performance of CLEF and its robustness to the choice of $\alpha$. To make this explicit, we have now conducted an ablation (now added to Appendix D.8 and Tbl. S17), where $\alpha$ is varied across {.1, .2, .3, .4, .5}. We observed: (1) The model remains robust when $\alpha$ is within range .1–.3, with .007 standard deviation averaged across all tasks. (2) Increasing $\alpha$ beyond .3 results in a slight performance drop, indicating that over-compressing the weighting factor can lead to model collapse. (3) Interestingly, if we compare the performance with Tbl. 4, it can be observed that the performance when $\alpha=.5$ is similar to SimCLR, implying the weighting factor is over-compressed and becomes less effective. Overall, our results confirm that while $\alpha$ can affect the shape of the embedding space, it does not directly drive the performance gains. The main improvements in the performance are due to using clinically-guided relative risk scores rather than the exact margin offset defined by $\alpha$. The updated ablation study indicates that the value of $\alpha=.2$ which was used in this study is appropriate.
>
> **Effect of contribution of $\mathcal{L}^w$ versus $\mathcal{L}^d$**: To clarify on the functional roles of our 2 losses, we initially performed preliminary experiments on images, as a domain-agnostic starting point (results not shown for CIFAR100 under differently weighted loss). To further confirm our choice of having equal weight over $\mathcal{L}^w$ and $\mathcal{L}^d$, we evaluated 4 configurations: (1) $\mathcal{L}^w$, (2) $\mathcal{L}^d$, (3) default balanced weight $\mathcal{L}^w+\mathcal{L}^d$, and (4) unbalanced mixes with coefficients $\lambda=$ {5,2,.5,.2} applied to $\mathcal{L}^w+\lambda\mathcal{L}^d$. The results are now added to Tbl. S16 in the revised version. These results show that CLEF is not particularly sensitive to the loss weight ratio (with an averaged standard deviation of .016 across all tasks), and the 2 components provide complementary guidance. Using balanced weights consistently provides strong performance, confirming our initial motivational analysis.
>
> **Computational cost**: The proposed pretraining method does not add significant computational complexity compared to baseline SimCLR. Risk scores are computed once from clinical metadata before training for each patient. During pretraining, the soft contrastive weighting adds minimal overhead by only computing the similarity between risk scores. We have added the discussion in our revised version in Appendix D.

---

> ### Author Response · Authors · 2025-11-23
> **Response to Questions**
>
> **Apply to other biosignals**: In principle, our clinically-guided weighting is not specific to ECG. It can be applied to other biosignals as long as (1) subject- or recording-level metadata is available, and (2) a validated function exists that maps this metadata to a clinically meaningful risk or similarity score. The architecture and loss formulation do not depend on an ECG-specific structure. While the approach is conceptually extensible to modalities, such as EEG or PPG, our work focuses exclusively on single-lead ECG. Extending CLEF to additional biosignals is a promising direction for future research. In the revised paper, specifically section 2.2 and Appendix F, we clarify that CLEF should be viewed as a general framework for risk-guided contrastive learning, with ECG serving as a natural application due to the availability of SCORE2 and rich metadata in MIMIC-IV.
>
> **Apply with additional risk scores**: We thank the reviewer for their suggestion of exploring additional risk scores. We would like to clarify that we do not claim that SCORE2 is the optimal risk score. Instead, the focus of our paper is to demonstrate the effectiveness of incorporating clinical guidance into contrastive learning. Verifying the method with other risk scoring systems is an interesting direction for future work. Indeed, we have now added Appendix F, discussing the choice of the SCORE2 risk score, its limitations, and considerations when selecting risk scores for future applications of CLEF. We believe investigating additional risk scores represents a future research direction.
>
> **Clinical interpretability of embeddings**: We have now visualised the latent space produced by CLEF using t-SNE visualisation ([anonymised link to the figure](https://anonymous.4open.science/r/image-1626/tsne_risk_score.png)). The visualization reveals a coarse, continuous distribution of risk scores across the embedding space, ranging from higher risk scores in the upper-left corner to lower risk scores in the lower-right corner. This demonstrates that CLEF effectively guides the model to produce embedding spaces that reflect the input risk score. There are no distinct clusters, which we believe is as expected because: (i) risk scores are numerical, implying a continuous distribution rather than discrete categories; (ii) Multiple pathologies can lead to similar risk scores but have varying effects on the ECG (e.g., acute hypertension vs. chronic hypertension alongside left ventricular hypertrophy); and (iii) Missing metadata introduces imprecision in the risk scores, contributing to less distinct boundaries in the embedding space.
>
> **Model performance under domain shift**: Although we do not explicitly quantify domain shift, our experimental design directly tests generalization across substantial shifts in acquisition device, sampling rate, patient population, and clinical context. CLEF is pretrained exclusively on MIMIC-IV-ECG (clinical 12-lead, 500 Hz), yet we evaluate on seven downstream datasets sourced from different institutions, countries, and hardware setups. In particular, (1) we use wearable ECG (Icentia11K) which differs most strongly from our pretraining dataset, with a different sampling rate (250 Hz), different demographic composition, and continuous ambulatory recordings rather than 10-second clinical ECGs. All three CLEF variants achieve performance comparable to or exceeding strong baselines on rhythm and beat classification tasks, demonstrating robustness under severe domain shift. (2) We use emergency-department and outpatient datasets (e.g., MC-MED, Aurora BP): These contain patient populations and collection protocols that differ substantially from MIMIC-IV. CLEF again maintains consistent improvements across classification and regression tasks. (3) Multi-center diagnostic datasets (PTB-XL, Chapman): These datasets reflect different geographic and institutional distributions, different patient mixes, and different ECG acquisition systems. CLEF continues to outperform or match the strongest foundation model baselines. Overall, CLEF is not limited to the MIMIC domain, and strong performance persists across datasets that differ from the pretraining distribution along multiple axes, including equipment type, sampling rate, demographic distribution, and clinical context.

---

### Author Response · Authors · 2025-11-23
**Summary of Revisions and Response to Reviewers (Part I)**

We sincerely appreciate the time and effort the Reviewers and Area Chairs devoted to evaluating our submission. We have made comprehensive revisions based on the insightful comments to strengthen the manuscript.

We are encouraged that the reviewers acknowledged the strengths of our work:

- **Novelty:** Our submission proposes a novel framework for electrocardiogram (ECG) foundation models (by ziBe).

- **Technical Contribution:** The idea of weighting negative pairs using risk score-derived dissimilarities is elegant and well-motivated, directly connecting the latent geometry of embeddings to real-world clinical semantics (by YTHd). The method is technically neat and conceptually coherent, converting clinical dissimilarity into a training signal (by ziBe).

- **Clinical Motivation:** The problem orientation is concrete and clinically motivated, targeting label scarcity and weak cross-device/population generalisation in single-lead ECG by injecting domain knowledge via routinely captured clinical metadata rather than dense manual labels (by ziBe). The model's connection to clinical reasoning adds conceptual clarity, which is often missing in current foundation model research (by YTHd).

- **Extensive Evaluation and Strong Empirical Results:** Strong empirical evaluation with broad benchmarking across diverse datasets and tasks, effectively bridging clinical and consumer-grade ECG domains (by YTHd). Performance gains are consistent against strong baselines, surpassing competitive self-supervised and supervised methods on standard metrics with improvements persisting across tasks, splits, and evaluation settings (by ziBe).

- **Rigorous experiments:** Comprehensive and rigorous experimentation with extensive evaluations and ablation studies across diverse datasets and clinical tasks (by SYZy). Robustness is carefully documented through multiple random seeds with confidence intervals and component-wise ablations that delineate the contribution of each design choice and indicate stable training behaviour (by ziBe).

- **Clear Presentation:** The framework (Figure 1) is well-illustrated, and the mathematical formulation of each component (Equations 5–7) is consistent and interpretable (by YTHd). The methodology and motivation are well articulated with intuitive explanations and clear visualisations, making the paper easy to follow even for non-domain experts (by SYZy).

---

> ### Author Response · Authors · 2025-11-29
> **Summary of Revisions and Response to Reviewers (Part II)**
>
> **Summary of Our Revisions**
>
> We have addressed major questions asked by the reviewers. We have conducted extensive additional experiments and provided detailed clarifications. Major updates incorporated into the revised manuscript include:
>
> **1. Additional experiments regarding important parameters of CLEF (Addressing concerns of Reviewers YTHd, ziBe)**: To demonstrate that our method is not hyperparameter sensitive, we expanded our evaluation to different parameter settings w.r.t $\alpha$ and $\tau$, and different contributions of weight components.
>
> - **Effect of parameter $\alpha$:** CLEF remains robust when $\alpha$ is within the range 0.1-0.3, with .007 standard deviation averaged across all tasks, supporting our choice of $\alpha=.2$. Over-compressing the weighting factor would make the pretrained model less effective. (Added in Appendix D.8, Table S17).
>
> - **Effect of parameter $\tau$:** Although the method is not overly sensitive to $\tau$, smaller $\tau$ ($\tau$={.07, .1}) consistently provide the strongest performance, indicating that lower $\tau$ enhances the discrimination between positives and negatives and is beneficial for learning clinical-informed ECG representations. (Added in Appendix D.8, Table S18).
>
> - **Difference loss weighting:** We extended our analysis to different weightings between the two losses. Empirical results show that while CLEF is not particularly sensitive to the loss weight ratio (averaged standard deviation is .016 across all tasks), using balanced weights consistently gives strong performance, supporting our choice. (Added in Appendix D.7, Table S16).
>
>
> **2. Additional clarification on the computational cost (Addressing the concern of Reviewer YTHd)** We added Appendix D.9, further clarifying that our proposed method added minimal computational overhead compared to the contrastive learning baseline, SimCLR.
>
> **3. Detailed clarification on the role of SCORE2 risk score (Addressing the concern of Reviewers YTHd, ziBe)**: We have added Appendix F to explain our choice of SCORE2. We clarified that the proposed framework supports any validated risk score, with the optimal choice depending on the specific application and available metadata. A corresponding note has been added to Section 2.2.
>
> **4. Clinical interpretability of embeddings (Addressing the concern of Reviewers YTHd, SYZy)**: We provided a t-SNE visualisation of CLEF's representation, demonstrating a coarse, continuous distribution of risk scores across the embedding space, confirming that CLEF effectively produces embeddings that reflect the input risk scores.
>
> **5. Further clarification on the rationale of handling missing data (Addressing the concern of Reviewer ziBe)**: We further clarified in Sections 2.2 and 2.3 that the missing metadata indicator $\mathbf{M}$ functions as a reliability adjustment: when metadata is incomplete and risk scores are unreliable, $\mathbf{M}$ moves the weighting closer to uniform SimCLR-like behaviour, preventing uncertain risk differences from unduly influencing the contrastive objective.
>
> We hope that these revisions and our detailed responses adequately address the concerns.
>
> **Code Release for Reproducibility**
>
> To encourage the reader and inspire reproducibility in the community, we have now provided an [anonymised codebase](https://anonymous.4open.science/r/ecg-foundation-model-A0E6) to supplement the paper.

---

### Meta-Review · Area_Chair_yd3W · 2025-12-23

**Summary:**

Summary: The paper proposes a contrastive learning framework for ECG data augmented with similarity scores that use domain-specific meta-data, particularly Systematic Coronary Risk Evaluation 2. Paper also demonstrates pretrained foundation models on 12-lead transferred to 1-lead, common to mobile health application. Across multiple datasets, their method demonstrates competitive performance. Clarifications to experiment design and method were added as part of rebuttal.

Reviewer concerns:
1. Robustness of results to hyperparameter tuning over the cost function and ablations: Multiple reviewers pointed out that the objective function does not consider the tradeoff of the dissimilarity alignment and contrastive losses ($\lambda$). In addition, there are several hyperparameters like the temperature $\tau$ of the contrastive loss, $\alpha$, which determines the baseline separation among negative samples that weren't tuned fully.

2. Reviewers are concerned that the claim of geometric alignment of the embeddings by augmenting using meta-data/risk score in the ECG case is not captured beyond impact on downstream performance, such as using visualizations or some other methods that demonstrates the impact on representations using the proposed domain-aligned weighting.

3. Reviewers raised the concern of the choice of meta-data, used in this case with ECG data, the clinically vetted SCORE2 risk score and its relation with downstream tasks. For instance, such meta data is collected based on 10 years of patient data, and the correlation with short-horizon downstream tasks, which are the main objects of evaluation are unclear.

4. The choice of representing missingness and its inconsistent impact on downstream performance was also brought up. The choice of not relying on imputation etc is not fully justified.

5. Overall reviewers also raised concerns about potential added fragility depending on the amount of missingness in incorporating the meta-data

6. Reviewer also pointed the emphasis on single-lead experimentation while using 12-lead data at pre-training.

**Reviewer Concerns:**

1. Based on reviewer concerns, authors added ablations to evaluating impact of tuning $\alpha$, $\lambda$, and $\tau$ on downstream performance. Authors note that certain values of $\alpha$ result in behavior akin to conventional simCLR (without weighting) but there exists a range of $\alpha$ values that maintains robust performance. Similarly, the choice of $\lambda$ does not significantly impact the objective.

2. Authors added t-sne visualization but rightfully pointed out the unreliability, limitations, and their lack of robustness to using such visualization techniques to analyze the claim of impact on geometry. My reading of reviewer comments is that reviewers are not tied to a specific method like t-sne or UMAP but want more evidence of the claim that beyond downstream performance, the alignment of embeddings, i.e., the geometry improves using the domain-inspired weighting scheme using metadata for ECG data.

3. Authors updated the draft to reflect the ablations including concerns about fragility in their response.

4. Authors correctly point to the importance of mobile health applications where single-lead tasks are prominent. In general this seems to be an issue with the presentation, since if the goal is to demonstrate utility of incorporating meta-data, especially on ECG representations, the downstream tasks should've mostly focused on tasks that learn and fine-tune on 12-lead data to demonstrate the hypothesis rather than adding an extra layer of complexity, even if that may be the motivation.

**Reviewer Scores:**

Based on the very polarized scores, my sense is that one of the two reviewers YTHd or ziBe might have considered increasing the score. Reviewer SYZy suggests that the contribution negates itself due to the emphasis on single-lead downstream applications. Authors have commented that the criticisms are unwarranted. I overall agree with the author's contention with the comments. However I am inclined to abstract out the following limitations that in my view remain unaddressed: i) if t-sne and umap are imperfect solutions to evaluate the claim of geometric alignment of learned embeddings, authors should either not claim it, and only emphasize downstream performance, or show alternative measures that provide evidence of the claim. ii) there is a confusing jump due to the application focus on single-lead ECG while the contribution emphasizing a general framework of "clinically aligning" pretrained embeddings, with an instantiation on ECG data. I'd say if this is the goal, then more application areas need to be presented, and if the downstream performance on consumer-wearable data is the goal then the contribution should be contextualized better for that application. Overall my sense is the presentation of the paper is lacking these nuances and remain unaddressed in the updated version.

---

### Decision · Program_Chairs · 2026-01-26

Reject